# Non-stationary Online Learning for Curved Losses:
# Improved Dynamic Regret via Mixability

**Yu-Jie Zhang** [1]  **Peng Zhao** [2]  **Masashi Sugiyama** [1 3]

## Abstract

Non-stationary online learning has drawn much attention in recent years. Despite considerable progress, dynamic regret minimization has primarily focused on convex functions, leaving the functions with stronger curvature (e.g., squared or logistic loss) underexplored. In this work, we address this gap by showing that the regret can be substantially improved by leveraging the concept of *mixability*, a property that generalizes exp-concavity to effectively capture loss curvature. Let $d$ denote the dimensionality and $P_T$ the path length of comparators that reflects the environmental non-stationarity. We demonstrate that an exponential-weight method with fixed-share updates achieves an $\mathcal{O}(dT^{1/3}P_T^{2/3}\log T)$ dynamic regret for mixable losses, improving upon the best-known $\mathcal{O}(d^{10/3}T^{1/3}P_T^{2/3}\log T)$ result (Baby & Wang, 2021) in $d$. More importantly, this improvement arises from a simple yet powerful analytical framework that exploits the mixability, which avoids the Karush-Kuhn-Tucker-based analysis required by existing work.

## 1. Introduction

Non-stationary online learning, which investigates how to learn from sequences of data in dynamic environments, has attracted considerable attention recently (Besbes et al., 2015; Zhang et al., 2018; Cutkosky, 2020; Baby & Wang, 2021; Wei & Luo, 2021; Zhao et al., 2022; Li et al., 2024; Zhao et al., 2024). A standard formulation for online learning is the online convex optimization (OCO) (Hazan, 2016), which involves a $T$-round process between the learner and the envi-

ronment. At each round $t \in [T] := \{1, \dots, T\}$, the learner submits a prediction $\mathbf{w}_t \in \mathcal{W}$ from a convex set $\mathcal{W} \subseteq \mathbb{R}^d$, while the environment simultaneously generates a convex loss function $f_t : \mathcal{W} \to \mathbb{R}$. Then, the learner incurs a loss of $f_t(\mathbf{w}_t)$ and observes the function $f_t$ for model update. In non-stationary environments, the dynamic regret (Zinkevich, 2003), which compares the learner's prediction with a time-varying benchmark $\{\mathbf{u}_t\}_{t=1}^T$,

$$\text{D-REG}_T(\{\mathbf{u}_t\}_{t=1}^T) = \sum_{t=1}^T f_t(\mathbf{w}_t) - \sum_{t=1}^T f_t(\mathbf{u}_t), \quad (1)$$

is commonly used to evaluate algorithms and guide their design. The dynamic regret (1) is usually called the "universal" dynamic regret as it holds for all comparator sequences. The universal dynamic regret not only recovers static regret by $\mathbf{u}_t = \mathbf{u}^* \in \arg\min_{\mathbf{w}\in\mathcal{W}} \sum_{t=1}^T f_t(\mathbf{w})$ for all $t \in [T]$, but also provides the flexibility to compare against any time-varying benchmark suited to the environment.

Over the years, dynamic regret minimization has been extensively studied for convex losses. Zinkevich (2003) showed that online gradient descent (OGD) with step size $\eta > 0$ achieves a dynamic regret bound of $\mathcal{O}((1 + P_T)/\eta + \eta T)$, which adapts to the path length

$$P_T(\mathbf{u}_1, \dots, \mathbf{u}_T) = \sum_{t=2}^T \|\mathbf{u}_t - \mathbf{u}_{t-1}\|_2,$$

reflecting the non-stationarity of the environments. Once the path length is known, the step size can be optimally tuned to $\eta_* = \Theta(\sqrt{(1 + P_T)/T})$. This choice yields the minimax optimal dynamic regret bound of $\mathcal{O}(\sqrt{T(1 + P_T)})$. However, given the universality of the comparator sequence, the path length is typically unknown, making optimal tuning infeasible. This highlights a fundamental challenge in dynamic regret minimization: how to handle the uncertainty of non-stationary environments. Zhang et al. (2018) addressed this issue by proposing a two-layer method that performs a grid search over possible path lengths, achieving the minimax optimal bound without knowledge of $P_T$. Beyond the minimax rate, the path-length based dynamic regret bound has attracted lots of interest for achieving problem-dependent bounds that adapts to the inherent hardness of the

---

[1]RIKEN AIP, Japan [2]National Key Laboratory for Novel Software Technology, Nanjing University, China [3]The University of Tokyo, Japan. Correspondence to: Yu-Jie Zhang <yu-jie.zhang@riken.jp>, Peng Zhao <zhaop@lamda.nju.edu.cn>.

*Proceedings of the $42^{nd}$ International Conference on Machine Learning*, Vancouver, Canada. PMLR 267, 2025. Copyright 2025 by the author(s).

**Table 1:** Comparison on dynamic regret and other issues on typical curved loss functions. * denotes that the method is proper learning when $\mathcal{W}$ is an $L_\infty$ ball. † indicates that the time complexity can be improved to $\mathcal{O}(\log T)$ with additional logarithmic factors in the regret.

| Losses | Method | Regret Bound | Proper Learning | Time Complexity |
|---|---|---|---|---|
| 1-dim squared loss $(z-y)^2$ | Baby & Wang (2021, Theorem 1) | $\widetilde{\mathcal{O}}(1 + T^{1/3}P_T^{2/3})$ | Yes | $\mathcal{O}(T)^\dagger$ |
| | Corollary 1 | $\widetilde{\mathcal{O}}(1 + T^{1/3}P_T^{2/3})$ | Yes | $\mathcal{O}(T)$ |
| Least-squares regression $(\mathbf{w}^\top \mathbf{x} - y)^2$ | Baby & Wang (2022b, Theorem 5) | $\widetilde{\mathcal{O}}(d + d^{10/3}T^{1/3}P_T^{2/3})$ | Yes | $\mathcal{O}(T)^\dagger$ |
| | Corollary 2 | $\widetilde{\mathcal{O}}(d + dT^{1/3}P_T^{2/3})$ | No | $\mathcal{O}(T)$ |
| Logistic regression $\log(1 + e^{-y\mathbf{w}^\top \mathbf{x}})$ | Baby et al. (2023, Theorem 3.1) | $\widetilde{\mathcal{O}}(d + d^{10/3}T^{1/3}P_T^{2/3})$ | Yes | $\mathcal{O}(T)$ |
| | Corollary 3 | $\widetilde{\mathcal{O}}(d + dT^{1/3}P_T^{2/3})$ | No | poly($T$) |
| General exp-concave loss | Baby & Wang (2022a, Theorem 10) | $\widetilde{\mathcal{O}}(d + d^{10/3}T^{1/3}P_T^{2/3})$ | No (in general)* | $\mathcal{O}(T)^\dagger$ |
| | Theorem 3 | $\widetilde{\mathcal{O}}(d + dT^{1/3}P_T^{2/3})$ | Yes | — |

learning problem (Zhao et al., 2020; 2024), being parameter-free (Cutkosky, 2020), and for unconstrained cases (Jacobsen & Cutkosky, 2023; Zhang et al., 2023b).

Despite considerable progress with convex functions, learning with *curved losses* for dynamic regret minimization remains underexplored. In online learning, curvature is often captured by exp-concavity or strong convexity (Hazan et al., 2007), which imposes specific conditions on the lower bound of the Hessian matrix $\nabla^2 f_t(\mathbf{w})$ of the loss. Key examples of exp-concave losses include:

- squared loss; $f_t(\mathbf{w}) = (\mathbf{w}^\top \mathbf{x}_t - y_t)^2$;
- logistic loss: $f_t(\mathbf{w}) = \log(1 + \exp(-y_t\mathbf{w}^\top \mathbf{x}_t))$;

Here, $\mathbf{x}_t \in \mathbb{R}^d$ is the feature, and $y_t$ is the label for the classification or regression tasks. For static regret minimization, where the comparator is fixed over the time, the online Newton step (ONS) achieves a logarithmic regret bound of $\mathcal{O}(\frac{d}{\eta}\log T)$ for $\eta$-exp-concave functions, while OGD with step size $\eta = \Theta(1/t)$ ensures $\mathcal{O}(\frac{1}{\lambda}\log T)$ regret for $\alpha$-strongly convex functions (Hazan et al., 2007).

However, when it comes to the dynamic regret minimization problem, the task becomes extremely challenging. To our knowledge, even if $P_T$ were known, it would remain unclear how to tune the step size of ONS or OGD to achieve a faster dynamic regret rate than the convex case. A recent breakthrough was achieved by Baby & Wang (2021) and further extended in Baby & Wang (2022a;b); Baby et al. (2023). The key is to introduce an offline optimal reference sequence, the behavior of which can be further characterized through a careful analysis of the KKT condition. They then demonstrate that a strongly adaptive algorithm—one that guarantees low regret over any interval—can effectively track this offline sequence, leading to a tight dynamic regret bound. Specifically, their method achieves a dynamic regret of $\widetilde{\mathcal{O}}\big(\max\{d^{\frac{10}{3}}T^{\frac{1}{3}}P_T^{\frac{2}{3}}, d\}\big)$ for exp-concave losses.[1]

---
[1] Baby & Wang (2021; 2022a) defined the path length with the $L_1$ norm. Here, we express their results in terms of the $L_2$ norm by $\|\mathbf{x}\|_1 \le \sqrt{d}\|\mathbf{x}\|_2$ for better comparison.

Besides, a lower bound of $\Omega\big(\max\{d^{\frac{1}{3}}T^{\frac{1}{3}}P_T^{\frac{2}{3}}, d\log T\}\big)$ is established for the $d$-dimensional squared loss, which is strongly convex and therefore exp-concave. While the upper bound are nearly optimal in terms of $T$ (up to logarithmic factors), the $O(d^{\frac{10}{3}})$ dependence on dimensionality for exp-concave losses exhibits a significant gap compared with the lower bound. More importantly, the intricacy of the KKT-based analysis makes it unclear how to extend their framework to achieve improved dependence on the dimensionality $d$ or to extend it for general results, such as obtaining problem-dependent bounds as the convex cases.

**Our Results.** This work offers a new perspective on achieving fast rates in dynamic regret minimization with curved losses by leveraging *mixability* (Vovk, 2001; van Erven et al., 2012) of the loss function, a concept closely related to exp-concavity and fundamental in fast-rate static regret minimization. Our method is free from the involved KKT-based analysis and enjoys improved regret on several curved loss functions with better dependence on the dimensionality $d$.

In Section 3 and Section 4, we begin with the online prediction problem, which is a specific kind of OCO problem with the form $f_t(\mathbf{w}) = \ell(h(\mathbf{w}_t, \mathbf{x}_t), y_t)$, where $h$ is a predictive function and $\ell$ is a certain loss. When the loss $\ell$ is mixable (see Definition 2) and improper learning is allowed, we show that a continuous variant of Vovk's aggregating algorithm (Vovk, 2001) with fixed-share updates attains a nearly optimal dynamic regret bound of $\widetilde{\mathcal{O}}\big(\max\{dT^{\frac{1}{3}}P_T^{\frac{2}{3}}, d\}\big)$.

As shown in Table 1, our approach covers several important curved losses. For the 1-dimensional squared loss, it matches the dynamic regret bound of Baby & Wang (2021) under *proper learning*, where the learner's predictions always lie within the feasible decision set. It also extends to the least-squares and logistic regression, two key examples of exp-concave online learning. Since both losses are also mixable, our method improves upon the best-known results (Baby & Wang, 2022b; Baby et al., 2023) by achieving a more favorable dependence on the dimensionality $d$. Importantly, the dynamic regret bound is derived via a rela-

tively simple yet effective decomposition based on the mix loss, which circumvents the need for a technically involved analysis of the KKT conditions. Nevertheless, while our method offers improvements in regret, it relies on improper learning, in contrast to the proper learning methods of Baby & Wang (2022b) for the least-square regression and Baby et al. (2023), which deals with the generalized linear models.

In Section 5, we further demonstrate the flexibility of our approach by extending it to the general OCO with exp-concave losses. We show that by incorporating an additional projection step, one can achieve the nearly optimal dynamic regret of $\widetilde{\mathcal{O}}\big(\max\{dT^{\frac{1}{3}}P_T^{\frac{2}{3}}, d\}\big)$ under proper learning. While the projection introduces non-trivial computational challenges, our result shows that one can attain a nearly optimal bound via proper learning over arbitrary convex and bounded domain for general exp-concave losses, a guarantee previously known only for specific losses such as the squared and logistic losses (Baby & Wang, 2022b; Baby et al., 2023).

Finally, we note that our method is based on the continuous exponential-weight (EW) update, which offers sharper regret guarantees than gradient-based methods but are often computationally more demanding. For the squared losses and least-squares regression, we have narrowed this gap by showing that a practical implementation attains the same $\mathcal{O}(T)$ per-round cost as the follow-the-leading-history (FLH) procedure in Baby & Wang (2021, Fig. 2). While FLH can be further accelerated to $\mathcal{O}(\log T)$ per round via a geometric covering over the base-learners (Hazan & Seshadri, 2009), extending this speed-up to the EW framework remains open. More generally, EW methods are sometimes surprisingly effective in many challenging online learning problems. However, improving their computational efficiency remains non-trivial, as in the case of bandit convex optimization (Ito, 2020; Bubeck et al., 2021).

## 2. Preliminaries

This section introduces mixability (Vovk, 2001), a property of losses that ensures fast rates in both statistical and online learning. The concept was initially studied in the prediction with expert advice setting and was shown to yield a constant static regret bound (Vovk, 1998). Later, it has also proved useful for static regret minimization in online optimization (Vovk, 2001; van der Hoeven et al., 2018) and for excess risk minimization in statistical learning (van Erven et al., 2012). We introduce the basic concepts here. Readers are referred to Cesa-Bianchi & Lugosi (2006, Chapter 3) and van Erven et al. (2015) for more details.

### 2.1. Mixability and Exp-concavity

Before introducing mixability, we review the related concept of exp-concavity (Hazan et al., 2007).

**Definition 1** (Exp-concavity)**.** Let $\eta > 0$. A function $f : \mathcal{W} \to \mathbb{R}$ is $\eta$-exp-concave over $\mathcal{W}$ if the function $e^{-\eta f(\mathbf{w})}$ is concave for any $\mathbf{w} \in \mathcal{W}$. The condition is equivalent to

$$f\left(\mathbf{w}_{\mathsf{exp}}\right) \le -\frac{1}{\eta} \ln\left(\mathbb{E}_{\mathbf{w} \sim P}\big[e^{-\eta f(\mathbf{w})}\big]\right), \qquad (2)$$

for any distribution $P$ over $\mathcal{W}$ and $\mathbf{w}_{\mathsf{exp}} = \mathbb{E}_{\mathbf{w} \sim P}[\mathbf{w}]$.

The mixability (Vovk, 2001) is weaker than exp-concavity that only requires the existence of a model $\mathbf{w}_{\mathsf{mix}}$ that satisfies the inequality (2) instead of the specific formulation of $\mathbf{w}_{\mathsf{exp}}$.

**Definition 2** (Mixability)**.** Let $\eta > 0$. A loss function $f : \mathcal{W} \to \mathbb{R}$ is $\eta$-mixable if for any distribution $P$ over the $\mathcal{W}$, there *always exists* a mapping $h : P \mapsto \mathbf{w}_{\mathsf{mix}}$ such that

$$f(\mathbf{w}_{\mathsf{mix}}) \le -\frac{1}{\eta} \ln\left(\mathbb{E}_{\mathbf{w} \sim P}\left[e^{-\eta f(\mathbf{w})}\right]\right). \qquad (3)$$

The coefficient $\eta$ in exp-concavity and mixability essentially characterizes the curvature of the loss function. A larger $\eta$ corresponds to a stronger curvature, which in turn leads to better regret guarantees. Moreover, any $\eta$-mixable loss remains $\eta'$-mixable for any $0 < \eta' \le \eta$.

A comparison of (2) and (3) shows that the key distinction lies in how the prediction is constructed from the distribution $P$. Under exp-concavity, the prediction is explicitly defined as $\mathbf{w}_{\mathsf{exp}} = \mathbb{E}_{\mathbf{w} \sim P}[\mathbf{w}]$, whereas mixability merely requires the existence of some model $\mathbf{w}_{\mathsf{mix}}$. Therefore, exp-concavity is a special case of mixability; any $\eta$-exp-concave function over $\mathcal{W}$ is at least $\eta$-mixable over $\mathcal{W}$ because the mixability condition (3) holds with $\mathbf{w}_{\mathsf{mix}} = \mathbf{w}_{\mathsf{exp}}$.

### 2.2. Examples of Mixable Loss

Here are two important examples of mixable loss.

**Example 1** (Squared Loss)**.** For any $y_t \in [-B, B]$,

$$f_{\mathsf{sq}}(z, y_t) = (z - y_t)^2$$

is $1/(2B^2)$-mixable over $z \in \mathbb{R}$ while it is $1/(2(B+D)^2)$-exp-concave over $z \in [-D, D]$.

**Example 2** (Logistic Loss)**.** For any $y_t \in \{-1, 1\}$,

$$f_{\mathsf{lr}}(z, y_t) = \log\left(1 + \exp(-zy_t)\right)$$

is 1-mixable over $z \in \mathbb{R}$ while it is $\exp(-D)$-exp-concave over $z \in [-D, D]$.

These two losses are two important examples arising in online prediction where the learner sequentially predicts a label $y_t$ based on observed input data (Vovk & Zhdanov, 2008). To highlight the core ideas of our approach, we initially focus on this supervised setting in Section 3 and Section 4. We then extend our method to the general online convex optimization (OCO) framework in Section 5.

**Algorithm 1** Fixed-share For Continuous Space

---

**Input:** mixability coefficient $\eta$ and parameter $\mu \in [0, 1]$.

1: Initialize $\mathbf{w}_0 \in \mathcal{W}$ as any point in the domain $\mathcal{W}$ and $\widetilde{P}_1 = P_1 = \mathcal{N}(\mathbf{w}_0, I_d)$ as a Gaussian distribution.

2: **for** $t = 1, 2, \dots, T$ **do**

3:     The learner submits the prediction $z_t$ that satisfies

$$\ell(z_t, y_t) \leq -\frac{1}{\eta} \ln \left( \mathbb{E}_{\mathbf{u} \sim P_t} \left[ e^{-\eta f_t(\mathbf{u})} \right] \right), \quad (4)$$

    without having access to the loss function $f_t$.

4:     The learner observes the loss function $f_t$ for time $t$.

5:     The learner updates the distributions by

$$\widetilde{P}_{t+1}(\mathbf{u}) \propto P_t(\mathbf{u}) \exp(-\eta f_t(\mathbf{u})), \forall \mathbf{u} \in \mathbb{R}^d; \quad (5)$$

$$P_{t+1}(\mathbf{u}) = (1 - \mu)\widetilde{P}_{t+1}(\mathbf{u}) + \mu N_0(\mathbf{u}), \quad (6)$$

    where $N_0 = \mathcal{N}(\mathbf{w}_0, I_d)$ is a Gaussian distribution.

6: **end for**

---

# 3. Proposed Method for Online Prediction

This section focuses on the online prediction problem. We begin by a generic framework for learning with mixable losses in the setting, followed by its theoretical analysis and an equivalent formulation for implementation.

## 3.1. Problem Setup

Online prediction is a special case of OCO, where the learner proceeds with data pairs $(\mathbf{x}_t, y_t)$, with $\mathbf{x}_t \in \mathcal{X} \subset \mathbb{R}^d$ denoting the feature and $y_t \in \mathbb{R}$ the label. At each round $t$, the learner first observes $\mathbf{x}_t$ and then produces a prediction $z_t \in \mathbb{R}$. After that, th learner incurs a loss $\ell(z_t, y_t)$ and subsequently observes the label $y_t$ to update the prediction.

As a natural extension of prior work on static regret minimization (Rakhlin et al., 2015; Mayo et al., 2022), our goal is to minimize dynamic regret with respect to a sequence of time-varying models in a given hypothesis space $\mathcal{H} = \{\mathbf{x} \mapsto h(\mathbf{w}, \mathbf{x}) \mid \mathbf{w} \in \mathcal{W}\}$, where $h : \mathcal{W} \times \mathcal{X} \to \mathcal{Y}$ is a fixed predictive model and $\mathcal{W} \subset \mathbb{R}^d$ denotes the parameter space. For instance, in the case of linear prediction, we have $h(\mathbf{w}, \mathbf{x}) = \mathbf{w}^\top \mathbf{x}$. The dynamic regret is then defined as

$$\text{D-REG}_T(\{\mathbf{u}_t\}_{t=1}^T) = \sum_{t=1}^T \ell(z_t, y_t) - \sum_{t=1}^T \ell(h(\mathbf{u}_t, \mathbf{x}_t), y_t),$$

where $\mathbf{u}_1, \dots, \mathbf{u}_T \in \mathcal{W}$ are time-varying model parameters. Online prediction be cast as standard OCO by setting $f_t(\mathbf{u}) = \ell(h(\mathbf{u}, \mathbf{x}_t), y_t)$, though we allow improper learning, where the prediction $z_t$ may fall outside the hypothesis space $\mathcal{H}$ (Shalev-Shwartz & Ben-David, 2014, Remark 3.2). Besides, we have the following assumptions.

**Assumption 1** (Bounded Domain). The parameter space

$\mathcal{W} \subseteq \mathbb{R}^d$ is convex and compact with diameter at most $D$, i.e., $\|\mathbf{w} - \mathbf{w}'\|_2 \leq D$ for all $\mathbf{w}, \mathbf{w}' \in \mathcal{W}$.

**Assumption 2** (Smoothness). The function $f_t(\mathbf{w}) = \ell(h(\mathbf{w}, \mathbf{x}_t), y_t)$ is $\beta$-smooth for any $\mathbf{w} \in \mathbb{R}^d$ and $t \in [T]$.

**Assumption 3** (Mixability). The loss function $\ell(z, y)$ is $\eta$-mixable over $z \in \mathbb{R}^d$ for any $y \in \mathcal{Y}$.

## 3.2. Generic Framework

Our method is an exponential-weight method with the fixed-share update over the continuous space. Instead of maintaining a model parameter $\mathbf{w}_t \in \mathcal{W}$ and predict $z_t = h(\mathbf{w}_t, \mathbf{x}_t)$, we maintain a distribution $P_t$ of the model parameter over $\mathbb{R}^d$. The prediction $z_t$ is made based on $P_t$ that satisfies the mixability condition (4). Under Assumption 3 and according to the definition of mixability, a predictor satisfying (4) always exists. For the online prediction problem, Vovk (2001, Eq. (11) and (12)) or Cesa-Bianchi & Lugosi (2006, Proposition 3.3) provided a general optimization framework for constructing such predictors. In the cases of squared loss and logistic loss, closed-form expressions for the predictor $z_t$ are available. Further details are provided in Section 4.

The update procedure of the distribution $P_t$ is summarized in Algorithm 1, where we initialize it with a Gaussian distribution $P_1 = \mathcal{N}(\mathbf{w}_0, I_d)$ at the first iteration. At each iteration $t$, the distribution $P_{t+1}$ is obtained by the exponential-weight update (5), followed by the fixed-share step (6). Algorithm 1 attains the following dynamic regret bound. Its proof sketch is given in Section 3.3.

**Theorem 1.** *Under Assumption 1, 2 and 3, for any $\mathbf{u}_t \in \mathcal{W}$, Algorithm 1 with $\mu = 1/T$ ensures*

$$\text{D-REG}_T(\{\mathbf{u}_t\}_{t=1}^T) \leq \mathcal{O}\left( d \log T \cdot (1 + T^{\frac{1}{3}} P_T^{\frac{2}{3}}) \right),$$

*where $P_T = \sum_{t=2}^T \|\mathbf{u}_t - \mathbf{u}_{t-1}\|_2$ is the path length.*

For mixable losses, Theorem 1 shows an $\widetilde{O}(d + dT^{\frac{1}{3}} P_T^{\frac{2}{3}})$ dynamic regret bound, which improves the $\widetilde{\mathcal{O}}(d + d^{\frac{10}{3}} T^{\frac{1}{3}} P_T^{\frac{2}{3}})$ dynamic regret bound established based on the notion of exp-concavity (Baby & Wang, 2022a), by reducing the dependence on dimensionality from $\mathcal{O}(d^{\frac{10}{3}})$ to $\mathcal{O}(d)$. As will be clear in Section 4, the improvement holds for loss functions for the least-squares regression and the logistic regression, which are both mixable and exp-concave.

Notably, Algorithm 1 does not require prior knowledge of the path length $P_T$, which quantifies the environmental non-stationarity. At first glance, this might seem surprising since, in the convex setting, existing approaches typically require a two-layer online ensemble to handle the uncertainty of the unknown $P_T$ (Zhao et al., 2024). The key distinction lies in the fact that our algorithm maintains a *distribution* $P_t$ rather than a single prediction $\mathbf{w}_t$. This enables the use of

a Gaussian comparator $Q_t$ in our analysis, which automatically adapts to non-stationary environments by *virtually* adjusting its variance as shown in Section 3.3. From an algorithmic perspective, we further show in Section 3.4 that our fixed-share method can be interpreted as, and is actually algorithmically equivalent to, a two-layer online ensemble.

### 3.3. Analysis of Algorithm 1

This part provides a proof sketch for Theorem 1, analyzing the dynamic regret of Algorithm 1 by leveraging the notion of mixability. Our analysis avoids the reliance on the technically involved KKT conditions as used in Baby & Wang (2021) and the subsequential works (Baby & Wang, 2022a). An exception is Zhang et al. (2023a), who studied the online covariate shift adaptation problem using the logistic loss and analyzed its dynamic regret without requiring KKT conditions. However, their analysis is restricted to a specific kind of comparator, defined as the minimizer of an unknown risk function, and additionally assumes the realizability of the comparators, which is not required by our method.

*Proof Sketch.* Let $P \in \mathcal{P}$ be a probability distribution and $\mathcal{P}$ is a set of all measurable distributions. The core concept of our analysis is the mix loss $m_t : \mathcal{P} \to \mathbb{R}$

$$m_t(P) = -\frac{1}{\eta} \ln \left( \mathbb{E}_{\mathbf{u} \sim P} \left[ \exp \left( -\eta f_t(\mathbf{u}) \right) \right] \right).$$

Let $z_t$ be the final prediction at round $t$. The dynamic regret can be decomposed into three terms:

$$\text{D-Reg}_T \leq \underbrace{\sum_{t=1}^{T} \ell(z_t, y_t) - \sum_{t=1}^{T} m_t(P_t)}_{\text{(A) mixability gap}} \quad (7)$$

$$+ \underbrace{\sum_{t=1}^{T} m_t(P_t) - \sum_{t=1}^{T} \mathbb{E}_{\mathbf{u} \sim Q_t}[f_t(\mathbf{u})]}_{\text{(B) mixability regret}}$$

$$+ \underbrace{\sum_{t=1}^{T} \mathbb{E}_{\mathbf{u} \sim Q_t}[f_t(\mathbf{u})] - \sum_{t=1}^{T} f_t(\mathbf{u}_t)}_{\text{(C) comparator gap}}.$$

The first term is the mixability gap, which measures the gap between the original loss and the mix loss. The second term is the "mixability regret", which is the regret in terms of the mix loss $m_t(P_t)$ and the expected loss over the sequence $\{Q_t\}_{t=1}^{T}$. The third term is called the "comparator gap" as it is the gap between the expected loss and the loss of $\mathbf{u}_t$.

We note that the distribution $Q_t$ only appears in the analysis and can be arbitrarily chosen to make the bound as tight as possible. Specifically, we choose $Q_t = \mathcal{N}(\mathbf{u}_t, \sigma^2 I_d)$ as the Gaussian distribution with the mean $\mathbf{u}_t$ and covariance $\sigma^2 I_d$, where $\sigma > 0$ is a parameter to be specified later.

① **Bounding mixability gap.** By (4) in Algorithm 1, the mixability gap is non-positive. Therefore, we can upper bound it by 0 and proceed with the second and third terms.

② **Bounding mixability regret.** We establish the following lemma for the mixability regret when comparing against the time-varying distribution $\{Q_t\}_{t=1}^{T}$.

**Lemma 1.** *Let $Q_t = \mathcal{N}(\mathbf{u}_t, \sigma^2 I_d)$ be a Gaussian distribution. The update rule (5) and (6) with $\mu = 1/T$ ensures*

$$\sum_{t=1}^{T} m_t(P_t) - \sum_{t=1}^{T} \mathbb{E}_{\mathbf{u} \sim Q_t}[f_t(\mathbf{u})] \leq \frac{1}{\eta} \left( 2 + \text{KL}(Q_1 \| P_1) \right)$$

$$+ \frac{1}{\eta} \sum_{t=2}^{T} \int_{\mathbf{u} \in \mathbb{R}^d} (Q_t(\mathbf{u}) - Q_{t-1}(\mathbf{u})) \ln \left( \frac{1}{P_t(\mathbf{u})} \right) d\mathbf{u},$$

*where $\text{KL}(P \| Q) = \mathbb{E}_{\mathbf{u} \sim P}[\ln (P(\mathbf{u})/Q(\mathbf{u}))]$.*

Given the Gaussinality of the distribution $Q_t$ and the initial distribution $\widetilde{P}_1 = \mathcal{N}(\mathbf{w}_0, I_d)$, several mathematical manipulations presented in Appendix A.3 show that

$$\texttt{term (B)} \lesssim \frac{d \log(\eta T) \cdot P_T/\sigma + d\sigma^2 T + d \log(1/\sigma)}{\eta},$$

where $\lesssim$ hides constants independent of $d$, $\sigma$, $T$ and $\eta$.

③ **Bounding comparator gap.** Then, we move on to the comparator gap. Owing to the smoothness of $f_t$, we have

$$\mathbb{E}_{\mathbf{u} \sim Q_t}[f_t(\mathbf{u}) - f_t(\mathbf{u}_t)]$$
$$\leq \mathbb{E}_{\mathbf{u} \sim Q_t} \left[ \langle \nabla f_t(\mathbf{u}_t), \mathbf{u} - \mathbf{u}_t \rangle + \frac{\beta}{2} \| \mathbf{u} - \mathbf{u}_t \|^2 \right] = \frac{\beta d \sigma^2}{2},$$

where the equality is by the Gaussinality of the distribution $Q_t$. Therefore, we have

$$\texttt{term (C)} = \sum_{t=1}^{T} \left( \mathbb{E}_{Q_t}[f_t(\mathbf{u})] - f_t(\mathbf{u}_t) \right) \leq \frac{\beta d \sigma^2 T}{2}. \quad (8)$$

**Combining the results.** By combining the upper bounds for term (B) and term (C), we achieve

$$\text{D-Reg}_T \lesssim \frac{1}{\eta} \left( \frac{d \log(\eta T) \cdot P_T}{\sigma} + \eta dT\sigma^2 + d \log \left( \frac{1}{\sigma} \right) \right).$$

Since the parameter $\sigma$ only appears in the analysis, we can freely tune it to make the above dynamic regret bound tight.

- When $P_T \leq \sqrt{1/T}$, we choose $\sigma = \sqrt{1/T}$ to obtain $\text{D-Reg}_T \leq \mathcal{O}(d \log T)$.

- When $P_T \geq \sqrt{1/T}$, we set $\sigma = \Theta(P_T^{\frac{1}{3}} T^{-\frac{1}{3}})$ to obtain $\text{D-Reg}_T \leq \mathcal{O}\left( d \log T \cdot \left( 1 + T^{\frac{1}{3}} P_T^{\frac{2}{3}} \right) \right)$.

We complete the proof by combining the two cases. $\qquad \square$

**Remark 1** (Technical Discussion). The mixability-based analysis was first developed for the prediction with expert advice (PEA) problem (Vovk, 1998), where the prediction domain $\mathcal{W}$ is a finite set. In the PEA setting, our analysis with fixed-share updates closely relates to that of Cesa-Bianchi et al. (2012b), which established a dynamic regret bound for the fixed-share method with convex losses. Although their arguments also apply to exp-concave losses, the later extension (Cesa-Bianchi et al., 2012a) only achieved a switching regret bound, a specific case of dynamic regret where the comparator sequence is piecewise stationary. In the OCO setting, van der Hoeven et al. (2018, Lemma 1) analyzed the exponential-weight method but primarily focused on the mixability regret term with respect to an unspecified and fixed $Q$. Technically, our results not only extend prior work with time-varying distributions $Q_t$, but also establish a path-length bound by carefully selecting $Q_t$ and analyzing the fixed-share method in continuous spaces. Our work addresses a gap in the literature by providing dynamic regret guarantees for mixable losses in the OCO setting.

### 3.4. An Equivalent Implementation

The implementation of our method involves two key aspects: (i) how to update the distribution $P_t$ and (ii) how to predict $z_t$ based on $P_t$ satisfying (4). This subsection provides an equivalent update rule for $P_t$, which has a more explicit form for implementation and helps us to better understand the mechanism of the algorithm. The construction of $z_t$ will be discussed in Section 4.

The key observation is that the fixed-share Algorithm 1 with fixed parameter $\mu$ is essentially equivalent to the follow-the-leading-history-type (FLH-type) algorithm (Hazan & Seshadhri, 2009) as summarized in Algorithm 2. The equivalence was first made in the PEA problem (Adamskiy et al., 2016), and here we show the results also extend to the continuous space, which leads to a more clear way of updating distribution $P_t$. The FLH-type method is summarized in Algorithm 2 and illustrated in Figure 1.

Conceptually, our method is essentially an online ensemble method (Zhao et al., 2024) consisting of two parts:

- **Multiple Base-learners** run over different time intervals. For each time $i$, we will invoke a new base-learner $\mathcal{B}_i$ with the initial distribution $P_i^{(i)} = \mathcal{N}(\mathbf{w}_0, I_d)$. Then, at the following time $t > i$, the distribution is updated by the exponential-weight update (9).

- **Meta-learner** assigns a weight to each base-learner to aggregate their predictions. For the new base-learner $\mathcal{B}_{t+1}$, the weight is set to $\mu$ and for the existing base-learners, the weights are updated by the Hedge method over the "mix loss" as shown in (10). The distribution $P_{t+1}$ is obtained by a weighted combination of the distributions of the base-learners.

---

**Algorithm 2** Follow-the-Leading-History

**Input:** mixability coefficient $\eta$ and parameter $\mu \in [0, 1]$.
1: Invoke the first base-learner $\mathcal{B}_1$ with the initial decision distribution $P_1^{(1)} = \mathcal{N}(\mathbf{w}_0, I_d)$.
2: Initialize base-learner pool $\mathcal{H}_1 = \{\mathcal{B}_1\}$, set weight $p_1^{(1)} = 1$ and set $P_1(\mathbf{w}) = P_1^{(1)}(\mathbf{w})$.
3: **for** $t = 1, 2, \ldots, T$ **do**
4:     The learner submits the prediction $z_t$ that satisfies (4).
5:     The learner observes the loss function $f_t$ for time $t$.
6:     For all base-learner $\mathcal{B}_i \in \mathcal{H}_t$, we update by

$$P_{t+1}^{(i)}(\mathbf{u}) \propto P_t^{(i)}(\mathbf{u}) \exp(-\eta f_t(\mathbf{u})), \forall \mathbf{u} \in \mathbb{R}^d. \quad (9)$$

7:     Initialize a new base-learner $\mathcal{B}_{t+1}$ whose decision distribution is $P_{t+1}^{(t+1)} = \mathcal{N}(\mathbf{w}_0, I_d)$.
8:     Update the weight for each base-learner $\mathcal{B}_i \in \mathcal{H}_t$ by

$$\widetilde{p}_{t+1}^{(i)} \propto p_t^{(i)} \cdot \mathbb{E}_{\mathbf{u} \sim P_t^{(i)}}[\exp(-\eta f_t(\mathbf{u}))]. \quad (10)$$

9:     Update the weight for existing base-learner by

$$p_{t+1}^{(i)} = \begin{cases} (1-\mu) \cdot \widetilde{p}_{t+1}^{(i)} & \text{for } \mathcal{B}_i \in \mathcal{H}_t \\ \mu & \text{for } \mathcal{B}_{t+1}. \end{cases} \quad (11)$$

10:     Update the base-learner pool: $\mathcal{H}_{t+1} = \mathcal{H}_t \cup \{\mathcal{B}_{t+1}\}$.
11:     Obtain $P_{t+1}(\mathbf{u}) = \sum_{\mathcal{B}_i \in \mathcal{H}_{t+1}} p_{t+1}^{(i)} P_{t+1}^{(i)}(\mathbf{u})$.
12: **end for**

---

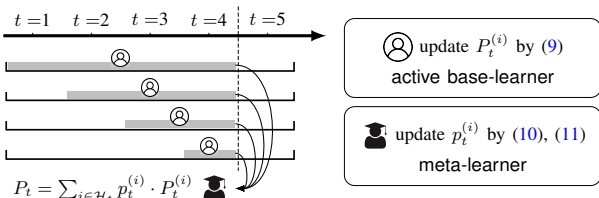

**Figure 1:** An illustration of Algorithm 2, an alternative implementation of the fixed share method.

The following theorem establishes the equivalence between Algorithm 1 and Algorithm 2.

**Theorem 2.** *The sequence of distributions $\{P_t\}_{t=1}^{T}$ returned by the fixed-share algorithm (Algorithm 1) is identical to that of the FLH-type algorithm (Algorithm 2) with the same sequence of input loss functions $\{f_t\}_{t=1}^{T}$ and the same parameter setting of $\mu$.*

An interesting connection between Algorithm 2 and the method of Baby & Wang (2021; 2022a) is that both approaches employ the FLH-type algorithm to bound dynamic regrets. However, their roles in the analyses differ substantially. In the previous work, the strongly adaptive regret guarantee of the FLH-type algorithm is pivotal for tracking the offline optimal sequence, whereas our analysis exploits a

fixed-share update formulation combined with mixability arguments, thereby offering a fresh perspective to achieve the nearly optimal dynamic regret bound. Furthermore, from a computational perspective, although Algorithm 2 aggregates a set of log-concave distributions—a task typically more computationally more intensive than updating a set of models $\mathbf{w}_t$ as in previous work—we show that for the squared loss, the distribution update can be performed in a closed form. This results in a computation cost that is of the same order as that of Baby & Wang (2021; 2022a) for these losses, but with an improved dependence on $d$.

# 4. Instantiation to Different Curved Losses

Now, we are ready to apply the generic algorithmic framework to different loss functions. We first consider the simple 1-dimensional squared loss for algorithmic illustration and then extend to the least-squares regression and logistic regression. We provide a comparison with the results by Baby & Wang (2021; 2022a) at the end of this section.

## 4.1. Instantiation to 1-Dimensional Squared Loss

The generic algorithmic framework can be directly applied to the 1-dimensional squared loss $f_t(z) = (z - y_t)^2$ as it is 2-smooth and $1/(2B^2)$ mixable for any $y_t \in [-B, B]$. Even better, we can construct the prediction $z_t^{\mathsf{mix}}$ with the greedy forecaster (Cesa-Bianchi & Lugosi, 2006). Specifically, given a distribution $P_t$ defined over $\mathcal{Z} = \mathbb{R}$, we can construct $z_t^{\mathsf{mix}}$ that satisfies the mixability condition (4) by

$$z_t^{\mathsf{mix}} = \left[ \frac{m_{\mathsf{sq}}(P_t, -B) - m_{\mathsf{sq}}(P_t, B)}{4B} \right]_B, \quad (12)$$

where we define $[z]_B = \min\{\max\{z, -B\}, B\}$ the function that clips the value $z$ to the interval $[-B, B]$. In the above, the mix loss is defined by

$$m_{\mathsf{sq}}(P_t, y) = -2B^2 \ln \left( \sum_{\mathcal{B}_i \in \mathcal{H}_t} p_t^{(i)} \cdot \mathbb{E}_{z \sim P_t^{(i)}} \left[ e^{-\frac{(z-y)^2}{2B^2}} \right] \right),$$

where $P_t^{(i)}$ is the distribution maintained by the base-learner $\mathcal{B}_i$ and $p_t^{(i)}$ is the weight from the meta-learner. A direct application of Theorem 1 leads to the following corollary.

**Corollary 1.** For the squared loss $f_t(z) = (z - y_t)^2$ with $y_t \in [-B, B]$, set $\mu = 1/T$ and $w_0 = 0$ in Algorithm 1. The prediction $z_t^{\mathsf{mix}}$ ensures

$$\sum_{t=1}^T f_t(z_t^{\mathsf{mix}}) - \sum_{t=1}^T f_t(u_t) \leq \widetilde{\mathcal{O}}(1 + T^{\frac{1}{3}} P_T^{\frac{2}{3}}),$$

where $P_t = \sum_{t=2}^T |u_t - u_{t-1}|$ is the path length.

**Implementations.** Since the squared loss is quadratic, the distribution $P_t^{(i)}$ is Gaussian, allowing for a closed-form update rule for both $P_t^{(i)}$ and $p_t^{(i)}$. Consequently, the

prediction $z_t^{\mathsf{mix}}$ can also be computed in a closed form. Since our method requires maintaining $t$ base learners at time $t$, the total time complexity for updating $P_{t+1}^{(i)}$ and $p_{t+1}^{(i)}$ and then aggregating them via (12) is $\mathcal{O}(t)$ per round. This computation cost is of the same order as the FLH method used by Baby & Wang (2021, Figure 2).

## 4.2. Instantiation to Least-Squares Regression

This subsection shows the fixed-share algorithm is also applicable to least-squares regression with the loss $f_t(\mathbf{w}) = (\mathbf{w}^\top \mathbf{x}_t - y_t)^2$. We assume that the feature vector $\mathbf{x}_t \in \mathbb{R}^d$ is bounded as $\|\mathbf{x}_t\|_2 \leq L$ and the label $y_t \in [-B, B]$. The basic observation is that the loss function $f_t$ is essentially the 1-dimensional squared loss $\ell_{\mathsf{sq}}(z, y_t) = (z - y_t)^2$ with $z = \mathbf{w}^\top \mathbf{x}_t$, which is $1/(2B^2)$-mixable. Therefore, for any distribution $P_t$ defined over $\mathbb{R}^d$, we can construct the prediction $z_t^{\mathsf{mix}}$ for the least-squares regression following the same rule as (12) with the mix loss $m(y, P_t)$ defined as

$$-2B^2 \ln \left( \sum_{\mathcal{B}_i \in \mathcal{H}_t} p_t^{(i)} \cdot \mathbb{E}_{\mathbf{w} \sim P_t^{(i)}} \left[ e^{-\frac{(\mathbf{w}^\top \mathbf{x}_t - y)^2}{2B^2}} \right] \right). \quad (13)$$

The aggregated prediction $z_t^{\mathsf{mix}}$ has the following guarantee for the least-squares regression.

**Corollary 2.** For the squared loss $f_t(\mathbf{w}) = (\mathbf{w}^\top \mathbf{x}_t - y_t)^2$ with $\|\mathbf{x}_t\|_2 \leq L$ and $y_t \in [-B, B]$, set $\mu = 1/T$ and $\mathbf{w}_0 = \mathbf{0}$. The mix prediction $z_t^{\mathsf{mix}}$ (13) ensures

$$\sum_{t=1}^T \ell_{\mathsf{sq}}(z_t^{\mathsf{mix}}, y_t) - \sum_{t=1}^T f_t(\mathbf{u}_t) \leq \widetilde{\mathcal{O}}(d + dT^{\frac{1}{3}} P_T^{\frac{2}{3}}),$$

where $P_T = \sum_{t=2}^T \|\mathbf{u}_t - \mathbf{u}_{t-1}\|_2$ is the path length of the comparator sequence with $\|\mathbf{u}_t\|_2 \leq D$.

**Implementations.** Similar to 1-dimensional squared loss, the distribution $P_t^{(i)}$ is a multivariate Gaussian distribution and we have a close form update rule for $z_t^{\mathsf{mix}}$. The method can be also implemented in $\mathcal{O}(t)$ time for each iteration $t$.

## 4.3. Instantiation to Logistic Regression

Given that logistic loss $\ell_{\mathsf{lr}}(z, y) = \log(1 + \exp(-yz))$ is 1-mixable for any $z \in \mathbb{R}^d$ and $y \in \{-1, 1\}$, the generic algorithmic framework is also applicable here as the case of least squares. Specifically, for any given distribution $P_t$ over $\mathbb{R}^d$, one can construct the mix prediction by

$$z_t^{\mathsf{mix}} = \sigma^{-1} \left( \sum_{\mathcal{B}_i \in \mathcal{H}_t} p_t^{(i)} \cdot \mathbb{E}_{\mathbf{w} \sim P_t^{(i)}} [\sigma(\mathbf{w}^\top \mathbf{x}_t)] \right), \quad (14)$$

where $\sigma(z) = 1/(1 + \exp(-z))$ is the sigmoid function and $\sigma^{-1}(z) = \ln \left( \frac{z}{1-z} \right)$ is the inverse function of the sigmoid. We have the following corollary for the logistic loss case.

**Corollary 3.** For the logistic loss $f_t(\mathbf{w}) = \log(1 + \exp(-y\mathbf{w}^\top \mathbf{x}))$ with $\|\mathbf{x}\|_2 \leq L$ and $y_t \in \{+1, -1\}$, set $\mu = 1/T$ and $\mathbf{w}_0 = \mathbf{0}$. The mix prediction $z_t^{\mathsf{mix}}$ ensures

$$\sum_{t=1}^T \ell_{\mathsf{lr}}(z_t^{\mathsf{mix}}, y_t) - \sum_{t=1}^T f_t(\mathbf{u}_t) \leq \widetilde{\mathcal{O}}(d + dT^{\frac{1}{3}} P_T^{\frac{2}{3}}),$$

where $P_T = \sum_{t=2}^T \|\mathbf{u}_t - \mathbf{u}_{t-1}\|_2$ is the path length of any comparator sequence such that $\|\mathbf{u}_t\|_2 \leq D$.

**Implementations.** For the logistic loss, the main computation cost is to calculate the term $\mathbb{E}_{\mathbf{u} \sim P_t^{(i)}}[\sigma(\mathbf{w}^\top \mathbf{x}_t)]$, which appears both in the weight update (10) and final prediction (14). We do not have a closed-form expression for this term. However, we can exploit the log-concavity of the density of $P_t^{(i)}$ and apply a sampling technique to facilitate computation in polynomial time (Foster et al., 2018, Appendix B). To further accelerate the computation, one might consider using the technique of Jézéquel et al. (2021), which approximates the logistic loss using a second-order surrogate. In such a case, $P_t^{(i)}$ is Gaussian and sampling can be performed efficiently. However, due to the two-layer structure of the method, it is challenging to directly extend the method here. We leave the computation cost for the logistic loss as a future work.

### 4.4. More Comparison with Related Work

In this subsection, we provide a detailed comparison between our results and best-known prior results, with a particular emphasis on the three main cases of the curved loss.

- For the 1-dimensional squared loss, our result matches the nearly optimal guarantee as Baby & Wang (2021, Theorem 1) and both methods are proper. However, our analysis does not rely on the KKT condition.

- For the least-squares regression and logistic regression, we improve the best-known results of $\widetilde{\mathcal{O}}(d + d^{\frac{10}{3}} T^{\frac{1}{3}} P_T^{\frac{2}{3}})$ in Baby & Wang (2022a;b) and Baby et al. (2023) with better dependence on $d$. For these two cases, Algorithm 1 is inherently improper, as its predictions extend beyond the linear function class. This is less favorable compared to the specific algorithms designed for the least-square regression (Baby & Wang, 2022b) and the logistic regression (Baby et al., 2023), which ensure proper learning.

We note that the path length in Baby & Wang (2021; 2022a) is defined via the $L_1$ norm. We present their results in the $L_2$ norm for better comparison through $\|\mathbf{x}\|_1 \leq \sqrt{d} \|\mathbf{x}\|_2$, $\forall \mathbf{x} \in \mathbb{R}^d$. The reliance on the $L_1$ norm in their analysis seems to stem from the inherent intricacy of handling KKT conditions. For the $L_1$ norm, the analysis in the $d$-dimensional case is closely connected to the 1-dimensional

case, as it can be proceeded in a coordinate-wise manner. In contrast, applying the KKT-based analysis to the $L_2$ norm without introducing additional dependence on $d$ remains unclear. Our mixability-based analysis offers a more direct way to obtaining path length bounds with the $L_2$ norm.

**Limitations of Algorithm 1 and Improvements.** Although Algorithm 1 achieves improved dynamic regret without relying on KKT-based analysis, there remain several directions for improvement. First, regarding the time complexity, our method matches the $\mathcal{O}(T)$ complexity of prior work for the squared losses. However, the KKT-based methods (Baby & Wang, 2021; 2022a) can reduce the complexity further to $\mathcal{O}(\log T)$ by leveraging a geometric covering of base-learners (Hazan & Seshadhri, 2009), at the expense of additional logarithmic factors in the regret. Incorporating this idea into the fixed-share update is a promising direction. Moreover, the guarantee provided by Algorithm 1 relies on the smoothness of the function $f_t$ and on allowing improper learning. In contrast, the smoothness condition is not required in Baby & Wang (2022a), and the works by Baby & Wang (2022b); Baby et al. (2023) eliminated the need of improper learning for online prediction with the linear model, where the loss has a specific form. In Section 5, we improve our method further by removing the assumptions on smoothness and improper learning, though the computational issue will become more challenging.

## 5. Extension to General OCO

This section extends our method to the general OCO without requiring a specific structure of the online function as in Section 3.1. Instead, we only assume the loss functions $f_t$ are exp-concave (and thus mixable) over the domain $\mathcal{W}$. By using a projection step and learning with a surrogate loss, our method achieves a nearly optimal regret bound under proper learning and is free from the smoothness assumption.

Besides Assumption 1 on the boundedness if the feasible domain $\mathcal{W}$, this section requires the following conditions.

**Assumption 4** (Exp-concave). The online function $f_t$ is $\eta$-exp-concave over the domain $\mathcal{W}$ for all $t \in [T]$.

**Assumption 5** (Bounded Gradient). The norm of the gradient of the loss function is bounded by $G$, i.e., $\|\nabla f_t(\mathbf{w})\|_2 \leq G$ for any $\mathbf{w} \in \mathcal{W}$ and $t \in [T]$.

A key obstacle in extending Algorithm 1 to the general OCO setting is that it requires the loss function to be mixable over $\mathbb{R}^d$, a condition often satisfied in online prediction but not in general OCO. In the latter case, exp-concavity is a more standard assumption, which only guarantees mixability over a bounded domain $\mathcal{W}$. This makes it challenging to construct a prediction $\mathbf{w}_t$ that has a negative mixability gap, i.e., $f_t(\mathbf{w}_t) \leq -\frac{1}{\eta} \ln \left( \mathbb{E}_{\mathbf{u} \sim P_t}[\exp(-\eta f_t(\mathbf{u}))] \right)$.

---

**Algorithm 3** Projected Fixed-share with Surrogate Loss

---

**Input:** exp-concavity coefficient $\eta$, loss parameter $\gamma$ and fixed-share parameter $\mu \in [0, 1]$.

1: Initialize $\mathbf{w}_0 \in \mathcal{W}$ as any point in $\mathcal{W}$ and $\widetilde{P}_1 = P_1 = \mathcal{N}(\mathbf{w}_0, I_d)$ as a Gaussian distribution.
2: **for** $t = 1, 2, \ldots, T$ **do**
3:    The learner submits $\mathbf{w}_t = \mathbb{E}_{\mathbf{w} \sim P_t}[\mathbf{w}]$ and constructs the surrogate loss $\widetilde{f}_t$ as (18).
4:    For all $\mathbf{u} \in \mathbb{R}^d$, the learner updates the by

$$P'_{t+1}(\mathbf{u}) \propto P_t(\mathbf{u}) \exp(-\gamma \widetilde{f}_t(\mathbf{u})/2); \quad (15)$$

$$\widetilde{P}_{t+1}(\mathbf{u}) = \arg\min_{Q \in \mathcal{M}} \mathrm{KL}(Q \| P'_{t+1}), \quad (16)$$

   where $\mathcal{P}$ is a set of distributions defined as (19).
5:    Then, the distribution is updated by the fixed share

$$P_{t+1}(\mathbf{u}) = (1 - \mu)\widetilde{P}_{t+1}(\mathbf{u}) + \mu N_0(\mathbf{u}), \quad (17)$$

   where $N_0 = \mathcal{N}(\mathbf{w}_0, I_d)$ is a Gaussian distribution.
6: **end for**

---

To address this issue, we introduce two algorithmic modifications as summarized in Algorithm 3. Our approach remains an exponential-weight method with fixed-share updates, but the distribution $P_t$ is now updated with a surrogate loss (15), and a projection step (16) is incorporated.

**Surrogate Loss.** Let $\mathbf{g}_t = \nabla f_t(\mathbf{w}_t)$. Instead of using the original loss, we learn with the surrogate loss defined by

$$\widetilde{f}_t(\mathbf{w}) = \mathbf{g}_t^\top(\mathbf{w} - \mathbf{w}_t) + \frac{\gamma}{2}\big(\mathbf{g}_t^\top(\mathbf{w} - \mathbf{w}_t)\big)^2, \quad (18)$$

where the coefficient is defined as $\gamma = \min\{1/(4GD), \eta\}$. It is known that the regret in terms of the surrogate loss is always an upper bound for the original loss: $f_t(\mathbf{w}_t) - f_t(\mathbf{u}_t) \le \widetilde{f}_t(\mathbf{w}_t) - \widetilde{f}_t(\mathbf{u})$ for any $\mathbf{w}_t, \mathbf{u} \in \mathcal{W}$ (Hazan, 2016, Lemma 4.2) for exp-concave losses. Hence, it suffices for us to analyze the dynamic regret of the surrogate loss.

The surrogate loss plays a key role in eliminating the smoothness assumption. In Section 3.3, smoothness was used to control the comparator gap. Since the surrogate loss is quadratic, we can directly upper-bound this term by

$$\texttt{term (C)} \le \mathbb{E}_{\mathbf{u} \sim Q_t}[\widetilde{f}_t(\mathbf{u}) - \widetilde{f}_t(\mathbf{u}_t)] \le d\sigma^2\gamma\|\mathbf{g}_t\|_2^2/2,$$

where we choose $Q_t = \mathcal{N}(\mathbf{u}_t, \sigma^2 I_d)$. The above inequality naturally leads to a similar result to (8) in Section 3.3 when the norm of the gradient is bounded as Assumption 5.

**Projection Step.** The remaining challenge is that $\widetilde{f}_t$ is not mixable over $\mathbb{R}^d$, making it hard to bound the mixability gap. However, we show that such a negative mixability gap can still be ensured if $P_t$ lies in a carefully designed distribution set $\mathcal{M}$, as guaranteed by the projection step (16).

Specifically, we choose the set $\mathcal{M}$ as the family of all Gaussian mixtures whose component means and covariances are uniformly bounded. Let $\mathcal{S} \triangleq \big\{\Sigma \in \mathbb{S}_{++}^d \mid \frac{1}{T} \le \lambda_{\min}(\Sigma) \le \lambda_{\max}(\Sigma) \le 1\big\}$ be a set of symmetric positive-definite (SPD) matrices with bounded eigenvalues, where $\mathbb{S}_{++}^d$ is the set of all SPD matrices. Further denote by $\Theta \triangleq \mathcal{W} \times \mathcal{S}$ the admissible parameter space for the component Gaussians and let $\theta = (\mathbf{w}, \Sigma) \in \Theta$. We define the constraint set as

$$\mathcal{M} \triangleq \left\{\int_{\theta \in \Theta} \mathcal{N}(\mathbf{w}, \Sigma) \mathrm{d}\pi(\theta) \mid \pi \in \mathscr{P}(\Theta)\right\}, \quad (19)$$

where $\mathscr{P}(\Theta)$ is the set of all probability measures on $\Theta$. As detailed in Appendix C, the set is convex and closed in total variation, therefore a minimizer in the projection step always exists (Csiszár, 1975, Theorem 2.1). Moreover, since any $P \in \mathcal{M}$ has its mean in $\mathcal{W}$, we have $\mathbf{w}_t = \mathbb{E}_{P_t}[\mathbf{w}] \in \mathcal{W}$, which ensures a proper prediction.

Lemma 7 in Appendix C shows that one can ensure a negative mixability gap for the surrogate loss, i.e.,

$$\widetilde{f}_t(\mathbb{E}_{\mathbf{w} \sim P_t}[\mathbf{w}]) \le -\frac{2}{\gamma}\ln\big(\mathbb{E}_{\mathbf{u} \sim P_t}[\exp(-\gamma\widetilde{f}_t(\mathbf{u})/2)]\big),$$

which leads to the following guarantee for Algorithm 3.

**Theorem 3.** *Under Assumptions 1, 4 and 5, Algorithm 3 with $\mu = 1/T$ ensures*

$$\text{D-R\textsc{eg}}_T \le \mathcal{O}\Big(d\log T \cdot (1 + T^{\frac{1}{3}}P_T^{\frac{2}{3}})\Big),$$

*for any comparator sequence $\mathbf{u}_1, \ldots, \mathbf{u}_T \in \mathcal{W}$.*

The proof of Theorem 3 is presented in Appendix C. Theorem 3 shows that our method achieves a nearly optimal dynamic regret for the general OCO with exp-concave functions. Notably, the result holds under proper learning and is free from the smoothness assumption.

## 6. Conclusion

This paper presented a novel perspective on learning with curved loss functions in non-stationary environments. Building on the concept of mixability, we proposed a simple yet versatile framework that achieves an improved dynamic regret bound with better dependence on $d$. Our analytical framework also offers additional potential, as it allows the selection of different families of distributions $P_t$ and reference distributions $Q_t$ for the fixed-share method, possibly enabling the algorithm to adapt to various geometric properties of the space. From a computational perspective, we provided efficient implementations for the squared loss. A future direction is to develop computationally more efficient methods for the logistic loss and the general OCO setting.

## Acknowledgements

Peng Zhao was supported by NSFC (62361146852, 62206125) and the Xiaomi Foundation. MS was supported by the Institute for AI and Beyond, UTokyo. The authors would like to thank Yu-Xiang Wang for the insightful and helpful discussions. Yu-Jie Zhang thanks Zhiyuan Zhan and Yivan Zhang for their helpful discussions on the projection set. We also thank the reviewers for the valuable comments and suggestions, which helped improve this paper.

## Impact Statement

This paper presents work whose goal is to advance the field of Machine Learning. There are many potential societal consequences of our work, none which we feel must be specifically highlighted here.

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

# A. Omitted Proofs for Section 3

## A.1. Useful Lemmas

**Lemma 2.** *Assume the loss function $f_t(\mathbf{u})$ is convex and $\beta$-smooth. Let $P_t$ be the distribution returned by the fixed-share update* (6). *Then, for any $\mu \in [0, 1]$, we have $\ln P_t(\mathbf{u}) \leq \frac{d}{2} \ln \left( \frac{\eta \beta t}{2\pi} \right)$.*

*Proof of Lemma 2.* The lemma can be proved by the equivalence between the fixed-share-type method (Algorithm 1) and the FLH-type method (Algorithm 2). Specifically, according to Theorem 2, the distribution $P_t$ returned by the fixed-share update (6) can be written as

$$P_t(\mathbf{u}) = \sum_{\mathcal{B}_i \in \mathcal{H}_t} p_t^{(i)} P_t^{(i)}(\mathbf{u}), \quad \forall \mathbf{u} \in \mathbb{R}^d,$$

where $p_t^{(i)} \in [0, 1]$ is the weight such that $\sum_{\mathcal{B}_i \in \mathcal{H}_i} p_t^{(i)} = 1$. The distribution $P_t^{(i)}(\mathbf{u})$ is updated via (9) and can be further expressed as

$$P_t^{(i)}(\mathbf{u}) = \frac{\exp \left( - F_t^{(i)}(\mathbf{u}) \right)}{\int_{\mathbf{u} \in \mathbb{R}^d} \exp \left( -F_t^{(i)}(\mathbf{u}) \right) \mathrm{d}\mathbf{u}}, \forall \mathbf{u} \in \mathbb{R}^d \text{ and } \mathcal{B}_i \in \mathcal{H}_t. \tag{20}$$

In the above, we denote by $F_t^{(i)}(\mathbf{u}) = \frac{1}{2} \|\mathbf{u} - \mathbf{w}_0\|_2^2 + \eta \sum_{s=i}^{t-1} f_s(\mathbf{u})$. Let $\mathbf{v}_{t,i}^* = \arg\min_{\mathbf{u} \in \mathbb{R}^d} F_t^{(i)}(\mathbf{u})$. Given the loss function $f_t$ is $\beta$-smooth, the first-order optimality condition implies

$$F_t^{(i)}(\mathbf{u}) \leq F_t^{(i)}(\mathbf{v}_{t,i}^*) + \frac{1 + \eta \beta t}{2} \|\mathbf{u} - \mathbf{v}_{t,i}^*\|_2^2, \ \forall \mathbf{u} \in \mathbb{R}^d \text{ and } \mathcal{B}_i \in \mathcal{H}_t.$$

Then, the denominator in (20) can be lower bounded by

$$\int_{\mathbf{u} \in \mathbb{R}^d} \exp \left( - F_t^{(i)}(\mathbf{u}) \right) \mathrm{d}\mathbf{u} \geq \exp(-F_t^{(i)}(\mathbf{v}_{t,i}^*)) \cdot \int_{\mathbf{u} \in \mathbb{R}^d} \exp \left( -\frac{1 + \eta \beta t}{2} \|\mathbf{u} - \mathbf{v}_{t,i}^*\|_2^2 \right) \mathrm{d}\mathbf{u}$$

$$= \exp \left( - F_t^{(i)}(\mathbf{v}_{t,i}^*) \right) \cdot \left( \frac{2\pi}{\eta \beta t} \right)^{d/2}. \tag{21}$$

For all $\mathbf{u} \in \mathbb{R}^d$ and $\mathcal{B}_i \in \mathcal{H}_t$, plugging (21) into (20) yields

$$P_t^{(i)}(\mathbf{u}) \leq \exp \left( F_t^{(i)}(\mathbf{v}_{t,i}^*) - F_t^{(i)}(\mathbf{u}) \right) \cdot \left( \tfrac{\eta \beta t}{2\pi} \right)^{\frac{d}{2}} \leq \left( \tfrac{\eta \beta t}{2\pi} \right)^{\frac{d}{2}}.$$

where the last inequality is due to the optimality of $\mathbf{v}_{t,i}^*$. Given $P_t(\mathbf{u})$ is a weighted combination of $P_t^{(i)}(\mathbf{u})$ and $P_t^{(i)}(\mathbf{u}) \geq 0, \forall \mathbf{u} \in \mathbb{R}^d$, we have $\max_{\mathbf{u} \in \mathbb{R}^d} P_t(\mathbf{u}) \leq \max_{\mathcal{B}_i \in \mathcal{H}_t, \mathbf{u} \in \mathbb{R}^d} P_t^{(i)}(\mathbf{u}) \leq (\eta \beta t / (2\pi))^{\frac{d}{2}}$. We complete the proof by taking logarithm on both sides. □

**Lemma 3.** *Let $P = \mathcal{N}(\mathbf{u}_p, \sigma^2 I_d)$ be a Gaussian distribution with mean $\mathbf{u}_p \in \mathbb{R}^d$ and covariance matrix $\sigma^2 I_d$ and $Q = \mathcal{N}(\mathbf{u}_q, \sigma^2 I_d)$. Besides, define $N = \mathcal{N}(\mathbf{w}_0, \lambda^2 I_d)$. Then, let $\mathcal{W} = \{\mathbf{u} \in \mathbb{R}^d \mid P(\mathbf{u}) \geq Q(\mathbf{u})\}$. Suppose $\|\mathbf{u}_p - \mathbf{w}_0\|_2 \leq D$ and $\|\mathbf{u}_q - \mathbf{w}_0\| \leq D$. Then, we have*

$$\int_{\mathbf{u} \in \mathcal{W}} (P(\mathbf{u}) - Q(\mathbf{u})) \ln \left( \frac{1}{N(\mathbf{u})} \right) \mathrm{d}\mathbf{u} \leq \left( \frac{d \ln(2\pi\lambda^2)}{2} + \frac{D}{\lambda^2} \right) \|P - Q\|_1 + \frac{2d\sigma^2}{\lambda^2},$$

*where $\|P - Q\|_1 = \int_{\mathbf{w} \in \mathbb{R}^d} |P(\mathbf{u}) - Q(\mathbf{u})| \mathrm{d}\mathbf{u}$.*

*Proof of Lemma 3.* Let $\mathbf{z} = \mathbf{u} - \mathbf{u}_p$. We define the shifted distribution $P'(\mathbf{z}) = \mathcal{N}(\mathbf{0}, \sigma^2 I_d)$, $Q'(\mathbf{z}) = \mathcal{N}(\mathbf{u}_q - \mathbf{u}_p, \sigma^2 I_d)$ and $N' = \mathcal{N}(\mathbf{w}_0 - \mathbf{u}_p, \sigma^2 I_d)$. To prove the lemma, it is equivalent to provide an upper bound for

$$\int_{\mathbf{z} \in \mathcal{W}'} (P'(\mathbf{z}) - Q'(\mathbf{z})) \ln \left( \frac{1}{N'(\mathbf{z})} \right) \mathrm{d}\mathbf{z}, \tag{22}$$

where $\mathcal{W}' = \{\mathbf{z} \in \mathbb{R}^d \mid P'(\mathbf{z}) \geq Q'(\mathbf{z})\}$ is the shifted region. According to the definition of multivariate Gaussian distribution $N'(\mathbf{z}) = \frac{1}{\sqrt{(2\pi\lambda^2)^d}} \exp(-\frac{1}{2\lambda^2}\|\mathbf{z} - (\mathbf{w}_0 - \mathbf{u}_p)\|_2^2)$. The term (22) can be equally written as:

$$\underbrace{\frac{d}{2}\ln(2\pi\lambda^2)\int_{\mathbf{z}\in\mathcal{W}'}(P'(\mathbf{z}) - Q'(\mathbf{z}))\,\mathrm{d}\mathbf{z}}_{\text{TERM (A)}} + \underbrace{\frac{1}{2\lambda^2}\int_{\mathbf{z}\in\mathcal{W}'}(P'(\mathbf{z}) - Q'(\mathbf{z}))\cdot\|\mathbf{z} - (\mathbf{w}_0 - \mathbf{u}_p)\|_2^2\mathrm{d}\mathbf{z}}_{\text{TERM (B)}}.$$

Denote by $\|P' - Q'\|_1 = \int_{\mathbf{z}\in\mathbb{R}^d}|P'(\mathbf{z}) - Q'(\mathbf{z})|\mathrm{d}\mathbf{z}$. TERM (A) can be directly bounded by

$$\text{TERM (A)} \leq \frac{d\ln(2\pi\lambda^2)}{2}\|P' - Q'\|_1 = \frac{d\ln(2\pi\lambda^2)}{2}\|P - Q\|_1. \tag{23}$$

As for TERM (B), define a $d$-dimensional ball centered around $\mathbf{a} \in \mathbb{R}^d$ with radius $R > 0$ as $\mathcal{B}(\mathbf{a}, R) = \{\mathbf{z} \in \mathbb{R}^d \mid \|\mathbf{z} - \mathbf{a}\|_2 \leq R\}$. We can further partition the region $\mathcal{W}'$ into two parts $\mathcal{W}'_{\text{in}} = \mathcal{W}' \cap \mathcal{B}(\mathbf{w}_0 - \mathbf{u}_p, 2D)$ and $\mathcal{W}'_{\text{out}} = \mathcal{W}'/\mathcal{W}'_{\text{in}}$. We have the following decomposition:

$$\begin{aligned}
\text{TERM (B)} &= \frac{1}{2\lambda^2}\int_{\mathbf{z}\in\mathcal{W}'_{\text{in}}}(P'(\mathbf{z}) - Q'(\mathbf{z}))\cdot\|\mathbf{z} - (\mathbf{w}_0 - \mathbf{u}_p)\|_2^2\mathrm{d}\mathbf{z} \\
&\quad + \frac{1}{2\lambda^2}\int_{\mathbf{z}\in\mathcal{W}'_{\text{out}}}(P'(\mathbf{z}) - Q'(\mathbf{z}))\cdot\|\mathbf{z} - (\mathbf{w}_0 - \mathbf{u}_p)\|_2^2\mathrm{d}\mathbf{z} \\
&\leq \frac{2D^2}{\lambda^2}\|P - Q\|_1 + \frac{1}{2\lambda^2}\int_{\mathbf{z}\in\mathcal{W}'_{\text{out}}}(P'(\mathbf{z}) - Q'(\mathbf{z}))\cdot\|\mathbf{z} - (\mathbf{w}_0 - \mathbf{u}_p)\|_2^2\mathrm{d}\mathbf{z} \\
&\leq \frac{2D^2}{\lambda^2}\|P - Q\|_1 + \frac{2}{\lambda^2}\int_{\mathbf{z}\in\mathcal{W}'_{\text{out}}}P'(\mathbf{z})\cdot\|\mathbf{z}\|_2^2\mathrm{d}\mathbf{z},
\end{aligned} \tag{24}$$

where the first inequality is due to $\|\mathbf{z} - (\mathbf{w}_0 - \mathbf{u}_p)\|_2 \leq 2D$ for any $\mathbf{z} \in \mathcal{W}'_{\text{in}}$. The second inequality holds since $\|\mathbf{z} - (\mathbf{w}_0 - \mathbf{u}_p)\|_2 \leq \|\mathbf{z}\|_2 + D \leq 2\|\mathbf{z}\|_2$ for any $\mathbf{z} \in \mathcal{W}'_{\text{out}}$.

We can further bound the last term in (24) by

$$\frac{2}{\lambda^2}\int_{\mathbf{z}\in\mathcal{W}'_{\text{out}}}P'(\mathbf{z})\cdot\|\mathbf{z}\|_2^2\mathrm{d}\mathbf{z} \leq \frac{2}{\lambda^2}\int_{\mathbb{R}^d}P'(\mathbf{z})\|\mathbf{z}\|_2^2\mathrm{d}\mathbf{z} = \frac{2}{\lambda^2}\mathbb{E}_{X_i\sim\mathcal{N}(0,1)}\left[\sigma^2\sum_{i=1}^d X_i^2\right] = \frac{2d\sigma^2}{\lambda^2} \tag{25}$$

where the first inequality is due to the fact $\mathcal{W}'_{\text{out}} \subset \mathbb{R}^d$. The last second equality is because one can show that the random variable $\mathbf{z} \in \mathbb{R}^d$ follows the same distributions as $\mathbf{z} \sim \sum_{i=1}^d \sigma X_i \mathbf{e}_i$, where $X_i \overset{\text{i.i.d.}}{\sim} \mathcal{N}(0,1)$ and $\mathbf{e}_i \in \mathbb{R}^d$ is the one-hot vector with the $i$-th element being 1 and others 0. The last equality follows by $\mathbb{E}_{X_i\sim\mathcal{N}(0,1)}[X_i^2] = 1$. Plugging (25) into (24), we obtain

$$\text{TERM (B)} \leq \frac{D}{\lambda^2}\|P - Q\|_1 + \frac{2d\sigma^2}{\lambda^2}.$$

We complete the proof by combining the upper bound for TERM (A) and TERM (B). $\qquad\square$

## A.2. Proof of Lemma 1

*Proof of Lemma 1.* Recall the definition of mix loss

$$m_t(P) = -\frac{1}{\eta}\ln\left(\mathbb{E}_{\mathbf{u}\sim P}\left[e^{-\eta f_t(\mathbf{u})}\right]\right).$$

By the exponential weights update rule in (5), we know that

$$\ln\left(\frac{P_t(\mathbf{u})}{\widetilde{P}_{t+1}(\mathbf{u})}\right) = \ln\left(\frac{\mathbb{E}_{P_t}\left[e^{-\eta f_t(\mathbf{u})}\right]}{e^{-\eta f_t(\mathbf{u})}}\right) = \ln\left(\mathbb{E}_{P_t}\left[e^{-\eta f_t(\mathbf{u})}\right]\right) + \eta f_t(\mathbf{u}),$$

which implies $-\frac{1}{\eta}\mathbb{E}_{\mathbf{u}\sim Q_t}\left[\ln\left(\frac{P_t(\mathbf{u})}{\widetilde{P}_{t+1}(\mathbf{u})}\right)\right] = m_t(P_t) - \mathbb{E}_{\mathbf{u}\sim Q_t}[f_t(\mathbf{u})]$ by taking an expectation of both sides with respect to $\mathbf{u}\sim Q_t$ and rearranging the terms. Summing this equality over $t\in[T]$ gives:

$$\sum_{t=1}^{T} m_t(P_t) - \sum_{t=1}^{T}\mathbb{E}_{\mathbf{u}\sim Q_t}[f_t(\mathbf{u})]$$

$$= -\frac{1}{\eta}\sum_{t=1}^{T}\mathbb{E}_{\mathbf{u}\sim Q_t}\left[\ln\left(\frac{P_t(\mathbf{u})}{\widetilde{P}_{t+1}(\mathbf{u})}\right)\right]$$

$$= \frac{1}{\eta}\sum_{t=1}^{T}\left(\text{KL}(Q_t\|P_t) - \text{KL}(Q_t\|\widetilde{P}_{t+1})\right)$$

$$\leq \frac{1}{\eta}\text{KL}(Q_1\|P_1) + \frac{1}{\eta}\sum_{t=2}^{T}\left(\text{KL}(Q_t\|P_t) - \text{KL}(Q_{t-1}\|\widetilde{P}_t)\right)$$

$$= \frac{1}{\eta}\text{KL}(Q_1\|P_1) + \frac{1}{\eta}\sum_{t=2}^{T}\left(\text{KL}(Q_t\|P_t) - \text{KL}(Q_{t-1}\|P_t)\right) + \frac{1}{\eta}\sum_{t=1}^{T}\mathbb{E}_{\mathbf{u}\sim Q_{t-1}}\left[\ln\left(\frac{\widetilde{P}_t(\mathbf{u})}{P_t(\mathbf{u})}\right)\right], \tag{26}$$

where both the second and last equalities are by the definition of the KL-divergence. In (26), the gap between the KL-divergence terms can be expressed as

$$\frac{1}{\eta}\sum_{t=2}^{T}\left(\text{KL}(Q_t\|P_t) - \text{KL}(Q_{t-1}\|P_t)\right)$$

$$= \frac{1}{\eta}\sum_{t=2}^{T}\left(\mathbb{E}_{\mathbf{u}\sim Q_t}\left[\ln\left(\frac{1}{P_t(\mathbf{u})}\right)\right] - \mathbb{E}_{\mathbf{u}\sim Q_{t-1}}\left[\ln\left(\frac{1}{P_t(\mathbf{u})}\right)\right]\right)$$

$$= \frac{1}{\eta}\sum_{t=2}^{T}\int_{\mathbf{u}\in\mathbb{R}^d}(Q_t(\mathbf{u}) - Q_{t-1}(\mathbf{u}))\ln\left(\frac{1}{P_t(\mathbf{u})}\right)d\mathbf{u}, \tag{27}$$

where the first equality is due to the definition of the KL divergence such that $\text{KL}(Q\|P) = \mathbb{E}_{\mathbf{u}\sim Q}\left[\ln\left(Q(\mathbf{u})/P(\mathbf{u})\right)\right]$ and $\mathbb{E}_{\mathbf{u}\sim Q_t}[\ln Q_t(\mathbf{u})] = \mathbb{E}_{\mathbf{u}\sim Q_{t-1}}[\ln Q_{t-1}(\mathbf{u})]$ for Gaussian distribution with the same mean. Furthermore, according to the fixed-share update rule (6), we know that $P_t(\mathbf{u}) = \mu N_0(\mathbf{u}) + (1-\mu)\widetilde{P}_t(\mathbf{u})$. Consequently, the final term in (26) can be bounded as

$$\frac{1}{\eta}\ln\left(\frac{\widetilde{P}_t(\mathbf{u})}{P_t(\mathbf{u})}\right) = \frac{1}{\eta}\ln\left(\frac{\widetilde{P}_t(\mathbf{u})}{(1-\mu)\widetilde{P}_t(\mathbf{u}) + \mu N_0(\mathbf{u})}\right) \leq \frac{1}{\eta}\ln\left(\frac{1}{1-\mu}\right) \leq \frac{2}{T\eta}, \tag{28}$$

where the last term is by the setting $\mu = 1/T$. Plugging (27) and (28) into (26) yields

$$\sum_{t=1}^{T} m_t(P_t) - \sum_{t=1}^{T}\mathbb{E}_{\mathbf{u}\sim Q_t}[f_t(\mathbf{u})] \leq \frac{1}{\eta}\sum_{t=2}^{T}\int_{\mathbf{u}\in\mathbb{R}^d}(Q_t(\mathbf{u}) - Q_{t-1}(\mathbf{u}))\ln\left(\frac{1}{P_t(\mathbf{u})}\right)d\mathbf{u} + \frac{2 + \text{KL}(Q_1\|P_1)}{\eta},$$

which completes the proof. $\qquad\square$

### A.3. Proof of Theorem 1

*Proof.* The core concept in our analysis is the mix loss, which has been used to analyze static regret in the prediction-with-expert-advice problem (Vovk, 1998) and in online convex optimization (van der Hoeven et al., 2018). Here, we show the critical role of mix loss in analyzing universal dynamic regret. For a probability distribution $P\in\mathcal{P}$, the mixed loss $m_t : \mathcal{P}\to\mathbb{R}$ is defined as

$$m_t(P) = -\frac{1}{\eta}\ln\left(\mathbb{E}_{\mathbf{u}\sim P}\left[e^{-\eta f_t(\mathbf{u})}\right]\right), \tag{29}$$

where $\eta$ is the mixability parameter.

Then, the dynamic regret can be decomposed into three terms as follow

$$
\text{D-Reg}_T = \sum_{t=1}^{T} \ell_t(z_t, y_t) - \sum_{t=1}^{T} f_t(\mathbf{u}_t)
$$

$$
= \underbrace{\sum_{t=1}^{T} \ell_t(z_t, y_t) - m_t(P_t)}_{\text{TERM (A)}} + \underbrace{\sum_{t=1}^{T} m_t(P_t) - \sum_{t=1}^{T} \mathbb{E}_{\mathbf{u} \sim Q_t}[f_t(\mathbf{u})]}_{\text{TERM (B)}} + \underbrace{\sum_{t=1}^{T} \mathbb{E}_{\mathbf{u} \sim Q_t}[f_t(\mathbf{u})] - \sum_{t=1}^{T} f_t(\mathbf{u}_t)}_{\text{TERM (C)}}, \qquad (30)
$$

where we choose $Q_t = \mathcal{N}(\mathbf{u}_t, \sigma^2 I_d)$ as a Gaussian distribution with mean $\mathbf{u}_t \in \mathbb{R}^d$ and covariance matrix $\sigma^2 I_d$. In the above, the TERM (A) is usually called the *mixability gap*, measuring the difference between the online function and the mixed loss. We call the TERM (B) the *mixability regret*, measuring the difference between the mixed loss and the loss measured on the expected comparators distributions. The TERM (C) is called the *comparator gap*, measuring the difference between the expected comparators distributions and the true comparators.

In the following, we will analyze the upper bound of each term, respectively.

① **Bounding mixability gap.**  According to (4) in Algorithm 1, the mixability gap is at most 0.

② **Bounding mixability regret.**  Lemma 1 shows that the mixability is upper bounded by

$$
\text{TERM (B)} \leq \underbrace{\frac{1}{\eta} \sum_{t=2}^{T} \int_{\mathbf{u} \in \mathbb{R}^d} (Q_t(\mathbf{u}) - Q_{t-1}(\mathbf{u})) \ln\left(\frac{1}{P_t(\mathbf{u})}\right) d\mathbf{u}}_{\text{TERM (B-I)}} + \underbrace{\frac{2 + \text{KL}(Q_1 \| P_1)}{\eta}}_{\text{TERM (B-II)}}, \qquad (31)
$$

*Analysis for Term (B-I):* For upper bounding TERM (B-I), the main challenge lies in the unboundedness of the logarithmic term $\ln(1/P_t(\mathbf{u}))$. This issue arises because the probability density $P_t(\mathbf{u})$ can become arbitrarily small even with the fixed-share update. To address this issue, we decompose the space $\mathbb{R}^d$ into:

$$
\mathcal{W}_t^{(1)} = \{\mathbf{u} \in \mathbb{R}^d \mid Q_t(\mathbf{u}) < Q_{t-1}(\mathbf{u}) \text{ and } P_t(\mathbf{u}) > 1\};
$$
$$
\mathcal{W}_t^{(2)} = \{\mathbf{u} \in \mathbb{R}^d \mid Q_t(\mathbf{u}) > Q_{t-1}(\mathbf{u}) \text{ and } P_t(\mathbf{u}) < 1\},
$$

and the region $\overline{\mathcal{W}}_t = \mathbb{R}^d / \mathcal{W}_t^{(1)} \cup \mathcal{W}_t^{(2)}$. For points $\mathbf{u} \in \overline{\mathcal{W}}$, we have $(Q_t(\mathbf{u}) - Q_{t-1}(\mathbf{u})) \ln(1/P_t(\mathbf{u})) \leq 0$, implying that the integral term over $\overline{\mathcal{W}}_t$ is non-positive.

Denote by $\|Q_t - Q_{t-1}\|_1 = \int_{\mathbf{u} \in \mathbb{R}^d} |Q_t(\mathbf{u}) - Q_{t-1}(\mathbf{u})| d\mathbf{u}$ the total variation of the distributions $Q_t$ and $Q_{t-1}$. For the integral over region $\mathcal{W}_t^{(1)}$, We have

$$
\int_{\mathbf{u} \in \mathcal{W}_t^{(1)}} (Q_t(\mathbf{u}) - Q_{t-1}(\mathbf{u})) \ln\left(\frac{1}{P_t(\mathbf{u})}\right) d\mathbf{u} = \int_{\mathbf{u} \in \mathcal{W}_t^{(1)}} (Q_{t-1}(\mathbf{u}) - Q_t(\mathbf{u})) \ln(P_t(\mathbf{u})) d\mathbf{u}
$$
$$
\leq \frac{d}{2} \ln\left(\frac{\eta \beta t}{2\pi}\right) \cdot \|Q_t - Q_{t-1}\|_1, \qquad (32)
$$

where the inequality holds due to Lemma 2 such that $\ln P_t(\mathbf{u}) \leq \frac{d}{2} \ln\left(\frac{\eta \beta t}{2\pi}\right)$ for $\mathbf{u} \in \mathbb{R}^d$.

As for the integral over the region $\mathcal{W}_t^{(2)}$, we have $Q_t(\mathbf{u}) > Q_{t-1}(\mathbf{u})$ and $\mu N_0(\mathbf{u}) \leq P_t(\mathbf{u}) \leq 1$ due to the fixed-share update rule (6). Then, we can bound the integral term by

$$
\int_{\mathbf{u} \in \mathcal{W}_t^{(2)}} (Q_t(\mathbf{u}) - Q_{t-1}(\mathbf{u})) \ln\left(\frac{1}{P_t(\mathbf{u})}\right) d\mathbf{u} \leq \int_{\mathbf{u} \in \mathcal{W}_t^{(2)}} (Q_t(\mathbf{u}) - Q_{t-1}(\mathbf{u})) \ln\left(\frac{1}{\mu N_0(\mathbf{u})}\right) d\mathbf{u}
$$
$$
\leq \left(\frac{d \ln(2\pi)}{2} + D + \ln\left(\frac{1}{\mu}\right)\right) \|Q_t - Q_{t-1}\|_1 + 2d\sigma^2, \qquad (33)
$$

where the last line holds due to Lemma 3. We obtain the upper bound for TERM (B-I) by combining (32) and (33) and taking the summation from $t = 2$ to $T$:

$$\text{TERM (B-I)} \leq \frac{1}{\eta} \left( \frac{d}{2} \ln(\eta \beta T) + D + \ln T \right) \sum_{t=2}^{T} \|Q_t - Q_{t-1}\|_1 + \frac{2d\sigma^2 T}{\eta}. \tag{34}$$

The total variation of the distributions $Q_t$ and $Q_{t-1}$ can be further bounded by

$$\|Q_t - Q_{t-1}\|_1 \leq \sqrt{\frac{1}{2} \text{KL}(Q_t \| Q_{t-1})} = \frac{\|\mathbf{u}_t - \mathbf{u}_{t-1}\|_2}{\sigma}, \tag{35}$$

where the first inequality is due to the Pinsker's inequality and the last equality is due to the close form expression of KL divergence of two Gaussian distribution as shown by Lemma 10. Combining (34) and (35), we obtain

$$\text{TERM (B-I)} \leq \frac{1}{\eta} \left( \frac{d \ln(\eta \beta T)}{2} + D + \ln T \right) \frac{P_T}{\sigma} + \frac{2d\sigma^2 T}{\eta},$$

where $P_T = \sum_{t=2}^{T} \|\mathbf{u}_t - \mathbf{u}_{t-1}\|_2$ is the path length.

*Analysis for Term (B-II):* Since $Q_1 = \mathcal{N}(\mathbf{u}_1, \sigma^2 I_d)$ and $P_1 = \mathcal{N}(\mathbf{w}_0, I_d)$ are both Gaussian distributions, Lemma 10 in Appendix D shows

$$\text{TERM (B-II)} = \frac{2}{\eta} + \frac{1}{\eta} \text{KL}(Q_1 \| P_1) \leq \frac{1}{\eta} \left( 2 + d \ln \left( \frac{1}{\sigma} \right) + \frac{d\sigma^2}{2} + \frac{D^2}{2} \right). \tag{36}$$

Then, we obtain the upper bound for TERM (B) by combining the above results as

$$\text{TERM (B)} \leq \frac{1}{\eta} \left( \frac{d \ln(\eta \beta T)}{2} + D + \ln T \right) \frac{P_T}{\sigma} + \frac{d\sigma^2(4T+1)}{2\eta} + \frac{D^2 + 2}{\eta} + \frac{d}{\eta} \ln \left( \frac{1}{\sigma} \right). \tag{37}$$

③ **Bounding comparator gap.** Since the loss function is $\beta$-smooth, we have

$$f_t(\mathbf{u}) \leq f_t(\mathbf{u}_t) + \langle \nabla f_t(\mathbf{u}_t), \mathbf{u} - \mathbf{u}_t \rangle + \frac{\beta}{2} \|\mathbf{u} - \mathbf{u}_t\|^2.$$

Then, taking an expectation over $Q_t$ for both sides, we have

$$\mathbb{E}_{\mathbf{u} \sim Q_t}[f_t(\mathbf{u})] \leq f_t(\mathbf{u}_t) + \frac{\beta}{2} \mathbb{E}_{\mathbf{u} \sim Q_t} \|\mathbf{u} - \mathbf{u}_t\|^2 \leq f_t(\mathbf{u}_t) + \frac{\beta d\sigma^2}{2}.$$

which implies

$$\text{TERM (C)} = \sum_{t=1}^{T} \mathbb{E}_{\mathbf{u} \sim Q_t}[f_t(\mathbf{u})] - \sum_{t=1}^{T} f_t(\mathbf{u}_t) \leq \frac{\beta d\sigma^2 T}{2}. \tag{38}$$

Plugging (37) and (38) into the decomposition (30), we have:

$$\text{D-REG}_T \leq \frac{1}{\eta} \left( \frac{d \ln(\beta \eta T)}{2} + D + \ln T \right) \frac{P_T}{\sigma} + \left( \frac{(4 + \eta \beta)T + 1}{2\eta} \right) d\sigma^2 + \frac{D^2 + 2}{\eta} + \frac{d}{\eta} \ln \left( \frac{1}{\sigma} \right).$$

We can further bound the above regret bound by considering different value of $P_T$.

**Case 1:** $P_T \leq \sqrt{\frac{1}{T}}$. We can choose $\sigma = \sqrt{1/T}$ to obtain D-REG$_T \leq \mathcal{O}(d \log T)$.

**Case 2:** $P_T \geq \sqrt{\frac{1}{T}}$. We select $\sigma = P_T^{\frac{1}{3}} (DT)^{-\frac{1}{3}} \leq 1$ to obtain D-REG$_T \leq \mathcal{O}\left( d \log T \cdot T^{\frac{1}{3}} P_T^{\frac{2}{3}} + d \log T \right)$ □

### A.4. Proof of Theorem 2

*Proof of Theorem 2.* We denote by $P_t^{\text{fs}}$ the distribution maintained by Algorithm 1 and $P_t^{\text{flh}}$ the distribution maintain by Algorithm 2. To prove the theorem, it is equivalent to show $P_t^{\text{fs}} = P_t^{\text{flh}}$ for all $t \in [T]$. For the update rule of $P_t^{\text{fs}}$, we have

$$
\begin{cases}
\widetilde{P}_{t+1}^{\text{fs}}(\mathbf{u}) = \frac{P_t^{\text{fs}}(\mathbf{u}) \exp(-\eta f_t(\mathbf{u}))}{\mathbb{E}_{\mathbf{u} \sim P_t^{\text{fs}}}[\exp(-\eta f_t(\mathbf{u}))]}. \\
P_{t+1}^{\text{fs}}(\mathbf{u}) = (1-\mu)\widetilde{P}_{t+1}^{\text{fs}}(\mathbf{u}) + \mu N_0(\mathbf{u}).
\end{cases}
\tag{39}
$$

We show that the update rule of $P_t^{\text{flh}}$ is equivalent to the above update rule. Recall $P_t^{\text{flh}}(\mathbf{u}) = \sum_{\mathcal{B}_i \in \mathcal{H}_t} p_t^{(i)} P_t^{(i)}(\mathbf{u})$, where $p_t^{(i)}$ and $P_t^{(i)}$ is the weights and distribution of the $i$-th base-learner respectively. Let us define

$$
\widetilde{P}_{t+1}^{\text{flh}}(\mathbf{u}) = \sum_{\mathcal{B}_i \in \mathcal{H}_t} \widetilde{p}_{t+1}^{(i)} P_{t+1}^{(i)}(\mathbf{u}),
$$

where $\widetilde{p}_{t+1}^{(i)}$ and $P_{t+1}^{(i)}$ is defined as (10) and (9), respectively. We have

$$
\begin{aligned}
\widetilde{P}_{t+1}^{\text{flh}}(\mathbf{u}) &= \sum_{\mathcal{B}_i \in \mathcal{H}_t} \widetilde{p}_{t+1}^{(i)} \cdot P_{t+1}^{(i)}(\mathbf{u}) \\
&= \sum_{\mathcal{B}_i \in \mathcal{H}_t} \frac{p_t^{(i)} \cdot \mathbb{E}_{\mathbf{u} \sim P_t^{(i)}}[\exp(-\gamma f_t(\mathbf{u}))]}{\sum_{\mathcal{B}_i \in \mathcal{H}_t} p_t^{(i)} \cdot \mathbb{E}_{\mathbf{u} \sim P_t^{(i)}}[\exp(-\eta f_t(\mathbf{u}))]} \cdot \frac{P_t^{(i)}(\mathbf{u}) \exp(-\eta f_t(\mathbf{u}))}{\mathbb{E}_{\mathbf{u} \sim P_t^{(i)}}[\exp(-\eta f_t(\mathbf{u}))]} \\
&= \frac{\sum_{\mathcal{B}_i \in \mathcal{H}_t} p_t^{(i)} \cdot P_t^{(i)}(\mathbf{u}) \exp(-\eta f_t(\mathbf{u}))}{\sum_{\mathcal{B}_i \in \mathcal{H}_t} p_t^{(i)} \cdot \mathbb{E}_{\mathbf{u} \sim P_t^{(i)}}[\exp(-\eta f_t(\mathbf{u}))]} = \frac{P_t^{\text{flh}}(\mathbf{u}) \exp(-\eta f_t(\mathbf{u}))}{\mathbb{E}_{\mathbf{u} \sim P_t^{\text{flh}}}[\exp(-\eta f_t(\mathbf{u}))]}.
\end{aligned}
$$

Besides, we can further rewrite the distribution $P_{t+1}^{\text{flh}}$ as

$$
P_{t+1}^{\text{flh}}(\mathbf{u}) = \sum_{\mathcal{B}_i \in \mathcal{H}_{t+1}} p_{t+1}^{(i)} P_{t+1}^{(i)}(\mathbf{u}) = (1-\mu) \sum_{\mathcal{B}_i \in \mathcal{H}_t} \widetilde{p}_{t+1}^{(i)} P_{t+1}^{(i)}(\mathbf{u}) + \mu N_0(\mathbf{u}) = (1-\mu)\widetilde{P}_{t+1}^{\text{flh}}(\mathbf{u}) + \mu N_0(\mathbf{u}).
$$

Therefore, we can conclude that the distribution maintain by Algorithm 2 can be equivalently updated by

$$
\begin{cases}
\widetilde{P}_{t+1}^{\text{flh}}(\mathbf{u}) = \frac{P_t^{\text{flh}}(\mathbf{u}) \exp(-\eta f_t(\mathbf{u}))}{\mathbb{E}_{\mathbf{u} \sim P_t^{\text{flh}}}[\exp(-\eta f_t(\mathbf{u}))]}. \\
P_{t+1}^{\text{flh}}(\mathbf{u}) = (1-\mu)\widetilde{P}_{t+1}^{\text{fs}}(\mathbf{u}) + \mu N_0(\mathbf{u}),
\end{cases}
$$

which is identical to (39). We complete the proof by the setting $P_1^{\text{fs}} = P_1^{\text{flh}} = N_0$. □

## B. Omitted Details for Section 4

This part provides omitted details in Section 4, including the detailed implementation of our method for squared loss and proofs of the statements in this section.

### B.1. Implementations for Squared Loss

Given the squared loss is quadratic and the initial distribution is Gaussian, we have close form update rule for $P_t^{(i)}$ and $p_t^{(i)}$.

**Proposition 1.** *The distributions $P_{t+1}^{(i)}$ for the base-learner $\mathcal{B}_i$ are Gaussian distribution $\mathcal{N}(w_{t+1,i}, B^2\sigma_{t+1,i}^2)$ for all $t = 1, \ldots, T-1$, whose means and variance are given by*

$$
w_{t+1,i} = \frac{w_{t,i} + \sigma_{t,i}^2 y_t}{1 + \sigma_{t,i}^2} \quad \text{and} \quad \sigma_{t+1,i}^2 = \frac{\sigma_{t,i}^2}{\sigma_{t,i}^2 + 1}
$$

*As for the meta-algorithm, due to the Gaussinality of $P_t^{(i)}$, the update of the weight $p_t^{(i)}$ in (10) is given by*

$$
\widetilde{p}_{t+1}^{(i)} \propto \frac{1}{\sqrt{1 + \sigma_{t,i}^2}} \cdot \exp\left(-\frac{(w_{t,i} - y_t)^2}{2B^2(\sigma_{t,i}^2 + 1)}\right).
$$

*Proof of Proposition 1.* According to the update rule (9), we have

$$P_{t+1}^{(i)}(w) \propto P_t^{(i)}(w) \cdot e^{-\frac{1}{2B^2}(w-y_t)^2}$$

$$\propto \exp\left(-\frac{1}{2B^2}\left(\frac{(w-w_{t,i})^2}{\sigma_{t,i}^2} + (w-y_t)^2\right)\right)$$

$$\propto \exp\left(-\frac{1}{2B^2}\left(\left(\frac{1}{\sigma_{t,i}^2}+1\right)w^2 - 2\left(\frac{w_{t,i}}{\sigma_{t,i}^2}+y_t\right)w + \text{const}\right)\right).$$

This represents a Gaussian distribution with updated parameters:

$$\sigma_{t+1,i}^2 = \left(\frac{1}{\sigma_{t,i}^2}+1\right)^{-1} = \frac{\sigma_{t,i}^2}{\sigma_{t,i}^2+1},$$

$$w_{t+1,i} = \sigma_{t+1,i}^2\left(\frac{w_{t,i}}{\sigma_{t,i}^2}+y_t\right) = \frac{w_{t,i}+\sigma_{t,i}^2 y_t}{\sigma_{t,i}^2+1}.$$

Therefore, $P_{t+1}^{(i)}$ is Gaussian with mean $w_{t+1,i}$ and variance $B^2\sigma_{t+1,i}^2$, as claimed. $\square$

## B.2. Useful Lemmas

**Lemma 4** (Mixability of Squared Loss). *For any $y_t \in [-Y,Y]$, the squared loss function $\ell_{sq}(z,y_t) = \frac{1}{2}(z-y_t)^2$ is $\frac{1}{2Y^2}$-mixable over the decision space $\mathcal{Z} = \mathbb{R}$.*

*Proof of Lemma 4.* According to Vovk (2001), the squared loss is $\frac{1}{2Y^2}$-mixable over the decision space $\mathcal{Z} = [-Y,Y]$, i.e., for any $y \in [-Y,Y]$ and $P'$ over $[-Y,Y]$, there exists a prediction $z'$ such that

$$\ell_{sq}(z',y) \leq -2Y^2 \ln\left(\mathbb{E}_{w\sim P'}\left[e^{-\frac{1}{2Y^2}\ell_{sq}(w,y)}\right]\right). \tag{40}$$

To prove the lemma, we observe that for any distribution $P$ over $\mathbb{R}$ and any $y$ in the interval $[-Y,Y]$, the always exists a distribution $P'$ defined over $[-Y,Y]$ satisfies that

$$-2Y^2 \ln\left(\mathbb{E}_{z\sim P}\left[e^{-\frac{1}{2Y^2}\ell_{sq}(z,y)}\right]\right) \geq -2Y^2 \ln\left(\mathbb{E}_{z\sim P'}\left[e^{-\frac{1}{2Y^2}\ell_{sq}(z,y)}\right]\right),$$

where $P'$ is a modified distribution that shifts the probability mass of $P$ outside the interval $[-Y,Y]$ to the boundary points. Specifically, $P'$ retains the same density as $P$ within $(-B,B)$, while at the boundaries, it is defined as: $P'(Y) = P(Y) + \int_{[Y,\infty)} P(z)\mathbf{d}z$ and $P'(-Y) = P(-Y) + \int_{(-\infty,-Y]} P(w)\mathbf{d}w$. According to (40), there exists a prediction $z'$ such that

$$\ell_t(z',y) \leq -2Y^2 \ln\left(\mathbb{E}_{z\sim P'}\left[e^{-\frac{1}{2Y^2}\ell_{sq}(z,y)}\right]\right) \leq -2Y^2 \ln\left(\mathbb{E}_{z\sim P}\left[e^{-\frac{1}{2Y^2}\ell_{sq}(z,y)}\right]\right),$$

thereby indicating the mixability over $\mathbb{R}$. $\square$

## B.3. Proof of Corollary 1

*Proof of Corollary 1.* To prove the corollary, it is sufficient to show that the prediction (12) satisfies the mixability inequality (3). We show (3) is essentially the greedy forecaster (Cesa-Bianchi & Lugosi, 2006, Section 3.4) for squared loss, which always ensures non-positive mixability gap.

As shown by Lemma 4, the squared loss is $\frac{1}{2B^2}$-mixable over the decision space $\mathcal{Z} = \mathbb{R}$. Then, for any mixable loss, Proposition 3.3 of Cesa-Bianchi & Lugosi (2006) shows that the greedy forecaster defined by

$$z^{\mathsf{mix}} = \underset{z\in\mathbb{R}}{\arg\min}\ \underset{y\in[-B,B]}{\sup}\left\{\underbrace{\ell_{sq}(z,y) - m_{sq}(P,y)}_{\texttt{mixability gap}}\right\}, \tag{41}$$

always ensures non-negative mixability gap, where $m_{\sf sq}(P, y) = -2B^2 \ln \left( \mathbb{E}_{w \sim P} \left[ e^{-\frac{1}{2B^2} \ell_{\sf sq}(z,y)} \right] \right)$ is the mix loss. Furthermore, as shown by Lemma 3 of Vovk (2001), the mixability gap $\ell_{\sf sq}(z, y) - m_{\sf sq}(P, y)$ is a convex function for any $y \in \mathbb{R}$ give a fixed $z$. Therefore, the inner optimization problem in (41) always achieves its optimal value of $y = B$ or $y = -B$. Then, we can equivalently rewrite the above optimization problem as

$$\arg \min_{z \in \mathbb{R}} \sup_{y \in \{-B, B\}} \{ \ell_{\sf sq}(z, y) - m_{\sf sq}(P, y) \}. \tag{42}$$

Then, it is sufficient to show the optimal solution of (42) is given as (12). Since $m_{sq}(P, y)$ is constant when the value of $y$ is given, the optimization problem (42) can then be reformulated as:

$$\arg \min_{z \in \mathbb{R}^d} \max \{ (z - B)^2 - M_1, (z + B)^2 - M_2 \},$$

where we define $M_1 = m_{sq}(P, B)$ and $M_2 = m_{sq}(P, -B)$. Further let $g_1(z) = (z - B)^2 - M_1$ and $g_2(z) = (z + B)^2 - M_2$. The inner optimization problem admits the following closed-form solution:

$$h(z) = \max \{ g_1(z), g_2(z) \} = \begin{cases} g_1(z), & \text{if } z \leq z^\dagger, \\ g_2(z), & \text{otherwise}, \end{cases}$$

where $z^\dagger = \frac{M_2 - M_1}{4B}$. We now consider three separate cases on the behavior of $h(z)$:

- **When** $-B \leq z^\dagger \leq B$: For $z \in (-\infty, z^\dagger]$, we have $h(z) = g_1(z)$, which is decreasing since $z^\dagger \leq B$. For $z \in [z^\dagger, \infty)$, we have $h(z) = g_2(z)$, which is increasing because $z^\dagger \geq -B$. Therefore, $h(z)$ attains its minimum at $z = z^\dagger$.

- **When** $z^\dagger < -B$: For the range $z \in (-\infty, z^\dagger]$, we have $h(z) = g_1(z)$ remains decreasing, and the minimum in this interval is at $z = z^\dagger$. On $z \in (z^\dagger, \infty)$, the minimum of $h(z)$ is at $z = -B$ since $h(-B) = g_2(-B) = -M_2$. Under the condition $z^\dagger \leq -B$, one can verify that $h(-B) \leq h(z^\dagger)$, so the overall minimum is attained at $z = -B$.

- **When** $z^\dagger > B$ By symmetry to the previous case, the minimum of $h(z)$ occurs at $z = B$ using the same reasoning.

Combining all three cases yields $\arg \min_{z \in \mathbb{R}^d} h(z) = \left[ \frac{M_2 - M_1}{4B} \right]_B$, which completes the proof. $\qquad \square$

### B.4. Proof of Corollary 2

*Proof of Corollary 2.* This corollary can be proved following the same arguments as Corollary 1. Specifically, the mix loss for least-squares can be equivalently written as

$$m_{\sf ls}(P_t, y) = -2B^2 \ln \left( \mathbb{E}_{\mathbf{w} \sim P_t} \left[ e^{-\frac{1}{2B^2} (\mathbf{w}^\top \mathbf{x}_t - y)^2} \right] \right) = -2B^2 \ln \left( \mathbb{E}_{z \sim P_t^{\mathcal{Z}}} \left[ e^{-\frac{1}{2B^2} (z - y)^2} \right] \right).$$

where $P_t^{\mathcal{Z}}$ is the distribution over $\mathbb{R}$ induced by $P_t$. Then, the arguments in the proof of Corollary 1 shows that the greedy forecaster (41) constructed based on $m_{ls}(P, y)$ ensures

$$\ell_{\sf sq}(z_t^{\sf mix}, y) \leq -2B^2 \ln \left( \mathbb{E}_{z \sim P_t^{\mathcal{Z}}} \left[ e^{-\frac{1}{2B^2} (z - y)^2} \right] \right) = -2B^2 \ln \left( \mathbb{E}_{\mathbf{w} \sim P_t} \left[ e^{-\frac{1}{2B^2} (\mathbf{w}^\top \mathbf{x}_t - y)^2} \right] \right),$$

which indicates a non-positive mixability gap and one can ensures an $\mathcal{O}((\ln T)^{\frac{2}{3}} P_T^{\frac{2}{3}} T^{\frac{1}{3}})$ bound following the same argument as Theorem 1. $\qquad \square$

### B.5. Proof of Corollary 3

*Proof of Corollary 3.* We show the mixability gap is non-positive when we defined $z_t^{\sf mix}$ as (14). Let $\sigma(z) = 1/(1 + \exp(-z))$. This can be shown by a direct calculation. In the case when $y = +1$, we have

$$\ell_{\sf lr}(z_t^{\sf mix}, +1) = \log \left( 1 + \exp(-z_t^{\sf mix}) \right) = -\ln \left( \mathbb{E}_{\mathbf{w} \sim P_t} [\sigma(\mathbf{w}^\top \mathbf{x}_t)] \right) = -\ln \left( \mathbb{E}_{\mathbf{w} \sim P_t} [e^{-\ell_{\sf lr}(\mathbf{w}^\top \mathbf{x}_t, +1)}] \right).$$

For the case $y = -1$, we have

$$\ell_{\sf lr}(z_t^{\sf mix}, -1) = \log \left( 1 + \exp(z_t^{\sf mix}) \right) = \log \left( \frac{1}{1 - \mathbb{E}_{\mathbf{w} \sim P_t} [\sigma(\mathbf{w}^\top \mathbf{x}_t)]} \right) = -\ln \left( \mathbb{E}_{\mathbf{w} \sim P_t} [e^{-\ell_{\sf lr}(\mathbf{w}^\top \mathbf{x}_t, -1)}] \right).$$

Therefore, the mixability gap is exactly 0 for any $y_t \in \{+1, -1\}$. We complete the proof by the same argument in the proof of Theorem 1. $\qquad \square$

# C. Omitted Proofs for Section 5

### C.1. Properties of the Set of Distributions $\mathcal{M}$

This section provides several properties about the set (19) defined as

$$\mathcal{M} \triangleq \left\{ \int_{\theta \in \Theta} \mathcal{N}(\mathbf{w}, \Sigma) \mathrm{d}\pi(\theta) \mid \pi \in \mathscr{P}(\Theta) \right\}, \tag{43}$$

where $\Theta = \mathcal{W} \times \mathcal{S}$ and we denote by $\theta = (\mathbf{w}, \Sigma)$. In the above, $\mathcal{S} \triangleq \left\{ \Sigma \in \mathbb{S}_{++}^d \mid \frac{1}{T} \leq \lambda_{\min}(\Sigma) \leq \lambda_{\max}(\Sigma) \leq 1 \right\}$ is a set of symmetric positive-definite matrices with bounded eigenvalue and $\mathcal{W} \subset \mathbb{R}^d$ is a convex and compact domain. We present the following lemmas about the properties of the set.

**Lemma 5.** *The set $\mathcal{M}$ is convex and closed in topology of the total variation distance.*

*Proof of Lemma 5.* We first check the convexity of the set. Pick two distributions $P_{\pi_1} = \int_{\theta \in \Theta} \mathcal{N}(\mathbf{w}, \Sigma) \mathrm{d}\pi_1(\theta)$, $P_{\pi_2} = \int_{\theta \in \Theta} \mathcal{N}(\mathbf{w}, \Sigma) \mathrm{d}\pi_2(\theta)$ in the set $\mathcal{M}$ and let $\lambda \in [0, 1]$. Set $\pi_\lambda = \lambda \pi_1 + (1 - \lambda)\pi_2$. Because $\pi_\lambda$ is a still probability measure on $\Theta$, it belongs to $\mathscr{P}(\Theta)$. For every Borel set $A \subseteq \mathbb{R}^d$,

$$P_{\pi_\lambda}(A) = \int_\Theta \mathcal{N}(\mathbf{w}, \Sigma)(A) \, \pi_\lambda(d\theta) = \lambda P_{\pi_1}(A) + (1 - \lambda)P_{\pi_2}(A),$$

so $P_{\pi_\lambda} = \lambda P_{\pi_1} + (1 - \lambda)P_{\pi_2} \in \mathcal{M}$. Hence $\mathcal{M}$ is convex.

Then, we check the set is closed in topology of the total variation distance. Denote by

$$g(\mathbf{u}; \theta) = \frac{1}{\sqrt{(2\pi)^d |\Sigma|}} \exp\left( -\frac{1}{2}(\mathbf{u} - \mathbf{w})^\top \Sigma^{-1}(\mathbf{u} - \mathbf{w}) \right)$$

the density function of the Gaussian distribution $\mathcal{N}(\mathbf{w}, \Sigma)$ with $\theta = (\mathbf{w}, \Sigma)$. We begin by choosing $\{P_n\}_{n \geq 1}$ as an arbitrary sequence within $\mathcal{M}$ such that $P_n$ convergence to a distribution $P_*$ in total variable, i.e., $\|P_n - P_*\|_{\mathrm{TV}} \to 0$. Because each $P_n$ lies in $\mathcal{M}$ by definition, it can be written in mixture form as $P_n = \int_\Theta \mathcal{N}(\mu, \Sigma) \, d\pi_n(\theta)$ for some $\pi_n \in \mathscr{P}(\Theta)$.

Since the parameter set $\Theta = \mathcal{W} \times \mathcal{S}$ is compact, every family of mixing measures $\{\pi_n\}_{n \geq 1} \subset \mathscr{P}(\Theta)$ is tight. By Prokhorov's theorem, one can extract a subsequence $(\pi_{n_k})_{k \geq 1}$ of $\{\pi_n\}_{n \geq 1}$ and find a limit $\pi \in \mathscr{P}(\Theta)$ such that

$$\pi_{n_k} \Longrightarrow \pi_* \quad \text{in } \mathscr{P}(\Theta) \quad \text{(weak convergence)}.$$

We can also define

$$P_{n_k}(\mathbf{u}) = \int_\Theta g(\mathbf{u}; \theta) \mathrm{d}\pi_{n_k}(\theta) \quad \text{and} \quad P_{\pi_*}(\mathbf{u}) = \int_\Theta g_\theta(\mathbf{u}) \mathrm{d}\pi_*(\theta).$$

Since the map $\theta \mapsto g(\mathbf{u}; \theta)$ is continuous and bounded on the compact set $\Theta$, we have

$$\lim_{k \to \infty} P_{n_k}(\mathbf{u}) = P_{\pi_*}(\mathbf{u}) \text{ for all } \mathbf{u} \in \mathbb{R}^d. \tag{44}$$

In the next, we show the density $P_{n_k}(\mathbf{u})$ and $P_{\pi_*}(\mathbf{u})$ is upper bounded by an integrable function $\bar{g}(\mathbf{u})$. Specifically, let $[z]_+ \triangleq \max\{0, z\}$ for any $z \in \mathbb{R}$. For every $\mathbf{u} \in \mathbb{R}^d$ we have

$$|P_{n_k}(\mathbf{u})| = \int_\Theta g(\mathbf{u}; \theta) \, \mathrm{d}\pi_{n_k}(\theta) \leq \bar{g}(\mathbf{u}) \triangleq \left( \frac{T}{2\pi} \right)^{d/2} \exp\left[ -\frac{1}{2} \left[ \|\mathbf{u}\|_2 - R \right]_+^2 \right],$$

where $R \triangleq \sup_{\mathbf{w} \in \mathcal{W}} \|\mathbf{w}\|_2 < \infty$ because $\mathcal{W}$ is compact. The inequality follows from the bounds $I_d \preccurlyeq \Sigma^{-1} \preccurlyeq T I_d$ that hold for every $(\mathbf{w}, \Sigma) \in \Theta$. The same argument gives $|P_{\pi_*}(\mathbf{u})| \leq \bar{g}(\mathbf{u})$ for all $\mathbf{u} \in \mathbb{R}^d$. Therefore, we have

$$|P_{n_k}(\mathbf{u}) - P_{\pi_*}(\mathbf{u})| \leq 2\bar{g}(\mathbf{u}). \tag{45}$$

We note the dominating function $\bar{g}$ is integrable as it is constant on the ball $\mathcal{C} = \{\mathbf{u} \in \mathbb{R}^d : \|\mathbf{u}\|_2 \leq R\}$ and decays with a Gaussian tail outside $\mathcal{C}$, hence $\int_{\mathbb{R}^d} \bar{g}(\mathbf{u}) \, \mathrm{d}\mathbf{u} < \infty \forall \mathbf{u} \in \mathbb{R}^d$.

Combining the pointwise convergence (44) and the integrable domination (45), the dominated convergence theorem yields

$$\lim_{k \to \infty} \int_{\mathbb{R}^d} |P_{n_k}(\mathbf{u}) - P_{\pi_*}(\mathbf{u})| d\mathbf{u} = 0. \tag{46}$$

Thus $P_{n_k} \to P_{\pi_*}$ in $L^1(\mathbb{R}^d)$. Because two probability measures with densities $q_1(\mathbf{u}), q_2(\mathbf{u})$ satisfy

$$\|Q_1 - Q_2\|_{\mathrm{TV}} = \frac{1}{2} \int_{\mathbb{R}^d} |q_1(\mathbf{u}) - q_2(\mathbf{u})| \, d\mathbf{u},$$

relation (46) implies

$$\|P_{n_k} - P_{\pi_*}\|_{\mathrm{TV}} = \tfrac{1}{2} \|P_{n_k} - P_{\pi_*}\|_{L^1} \xrightarrow{k \to \infty} 0. \tag{47}$$

By assumption, the original sequence $\{P_n\}_{n \geq 1}$ converges in total variation to some limit $P_*$. Because total variation is a metric, limits are unique; therefore $P_* = P_{\pi_*}$. Finally, $P_{\pi_*}$ has already been written as the Gaussian mixture generated by the weak limit $\pi_*$ of the mixing measures, so $P_{\pi_*} \in \mathcal{M}$. Consequently, every TV-convergent sequence in $\mathcal{M}$ remains in $\mathcal{M}$, and the set is closed in the total-variation topology. $\qquad\square$

**Lemma 6.** *For any distribution $P_t \in \mathcal{M}$, we have $\max_{\mathbf{u} \in \mathbb{R}^d} \ln(P_t(\mathbf{u})) \leq \frac{d}{2} \ln\left(\frac{T}{2\pi}\right)$.*

*Proof of Lemma 6.* According to the definition of $\mathcal{M}$, we have $P_t = \int_{(\mathbf{w}, \Sigma) \in \Theta} \mathcal{N}(\mathbf{w}, \Sigma) d\pi_t(\theta)$ for some distribution $\pi_t$ on $\Theta$. Then, for any $\mathbf{u} \in \mathbb{R}^d$, the density can be bounded by

$$
\begin{aligned}
P_t(\mathbf{u}) &= \int_{(\mathbf{w}, \Sigma) \in \mathcal{W} \times \mathcal{S}} \frac{1}{\sqrt{(2\pi)^d |\Sigma|}} \exp\left(-\frac{1}{2}(\mathbf{u} - \mathbf{w})^\top \Sigma^{-1}(\mathbf{u} - \mathbf{w})\right) d\pi_t(\theta) \\
&\leq \int_{(\mathbf{w}, \Sigma) \in \mathcal{W} \times \mathcal{S}} \frac{1}{\sqrt{(2\pi)^d |\Sigma|}} d\pi_t(\theta) \\
&\leq \max_{(\mathbf{w}, \Sigma) \in \mathcal{W} \times \mathcal{S}} \frac{1}{\sqrt{(2\pi)^d |\Sigma|}} \leq \frac{1}{\sqrt{(2\pi/T)^d}}
\end{aligned}
$$

where the first inequality is due to the fact that $\exp(-z) \leq 1$ for any $z \geq 0$. The last inequality is due to the definition of $\mathcal{S} \triangleq \{\Sigma \in \mathbb{S}_{++}^d \mid \frac{1}{T} \leq \lambda_{\min}(\Sigma) \leq \lambda_{\max}(\Sigma) \leq 1\}$ and the fact that $\lambda_{\min}(\Sigma) \geq 1/T$ for any $\Sigma \in \mathcal{S}$. Then, we have $\ln(P_t(\mathbf{u})) \leq \ln\left(\frac{1}{\sqrt{(2\pi/T)^d}}\right) \leq \frac{d}{2} \ln\left(\frac{T}{2\pi}\right)$ for any $\mathbf{u} \in \mathbb{R}^d$, which completes the proof. $\qquad\square$

### C.2. Useful Lemmas

**Lemma 7.** *Let $\widetilde{f}_t(\mathbf{w}) = \mathbf{g}_t^\top(\mathbf{w} - \mathbf{w}_t) + \frac{\gamma}{2}\left(\mathbf{g}_t^\top(\mathbf{w} - \mathbf{w}_t)\right)^2$, where $\mathbf{w}_t = \mathbb{E}_{\mathbf{w} \sim P_t}[\mathbf{w}]$. Under Assumptions 1 and 5, we have*

$$\widetilde{f}_t(\mathbf{w}_t) \leq -\frac{2}{\gamma} \ln\left(\mathbb{E}_{\mathbf{w} \sim P_t}\left[\exp\left(-\frac{\gamma}{2}\widetilde{f}_t(\mathbf{w})\right)\right]\right)$$

*for any distribution $P_t \in \mathcal{M}$ when $\gamma \leq 1/(4GD)$ and $\mathbf{w}_t \in \mathcal{W}$.*

*Proof of Lemma 7.* To prove the lemma, it is sufficient to show $\mathbb{E}_{\mathbf{w} \sim P_t}[\exp(-\gamma \widetilde{f}_t(\mathbf{w})/2)] \leq \exp(-\gamma \widetilde{f}_t(\mathbf{w}_t)/2)$ for the distribution $P_t \in \mathcal{M}$. Given the definition of $\mathcal{M}$ (19), the distribution $P_t = \int_{(\mathbf{w}, \Sigma) \in \Theta} \mathcal{N}(\mathbf{w}, \Sigma) d\pi_t(\theta)$ is a Gaussian

mixture model, where $\pi_t$ is a distribution on $\Theta$ corresponding to $P_t$. Then, we have

$$
\begin{aligned}
\mathbb{E}_{\mathbf{w} \sim P_t}[\exp(-\gamma \widetilde{f}_t(\mathbf{w})/2)] &= \int_{(\mathbf{w},\Sigma) \in \Theta} \mathbb{E}_{\mathbf{u} \sim \mathcal{N}(\mathbf{w},\Sigma)}[\exp(-\gamma \widetilde{f}_t(\mathbf{w})/2)] \pi_t(d\theta) \\
&= \int_{(\mathbf{w},\Sigma) \in \Theta} \mathbb{E}_{\mathbf{u} \sim \mathcal{N}(\mathbf{w},\Sigma)} \left[ \exp \left( \frac{\gamma}{2} \mathbf{g}_t^\top (\mathbf{w}_t - \mathbf{u}) - \frac{\gamma^2}{4} \left( \mathbf{g}_t^\top (\mathbf{u} - \mathbf{w}_t) \right)^2 \right) \right] \pi_t(d\theta) \\
&\leq \int_{(\mathbf{w},\Sigma) \in \Theta} \exp \left( \frac{\gamma}{2} \mathbf{g}_t^\top (\mathbf{w}_t - \mathbf{w}) - \frac{\gamma^2}{4} \left( \mathbf{g}_t^\top (\mathbf{w} - \mathbf{w}_t) \right)^2 \right) \pi_t(d\theta) \\
&\leq \int_{(\mathbf{w},\Sigma) \in \Theta} \left( 1 + \frac{\gamma}{2} \mathbf{g}_t^\top (\mathbf{w}_t - \mathbf{w}) \right) \pi_t(d\theta) \\
&= 1 + \frac{\gamma}{2} \mathbf{g}_t^\top \left( \mathbf{w}_t - \int_{(\mathbf{w},\Sigma) \in \Theta} \mathbb{E}_{\mathbf{u} \sim \mathcal{N}(\mathbf{w},\Sigma)}[\mathbf{u}] \pi_t(d\theta) \right) = 1.
\end{aligned}
$$

where first inequality is due to the Gaussian exp-concavity as shown in Lemma 12 in Appendix D under the condition $\gamma/2 \leq 1/(5GD)$. The second inequality follows from the fact that $e^{z-z^2} \leq 1 + z$ for any $z \geq -\frac{2}{3}$. In our case, we set $z = -\frac{\gamma}{2} \mathbf{g}_t^\top (\mathbf{w}_t - \mathbf{w})$, which satisfies $z \geq -\frac{2}{3}$ under the condition on $\gamma$ and the boundedness $\|\mathbf{w}_t - \mathbf{w}\|_2 \leq D$ for all $\mathbf{w}_t, \mathbf{w} \in \mathcal{W}$. The last inequality is due to $\int_{(\mathbf{w},\Sigma)} \mathbb{E}_{\mathbf{u} \sim \mathcal{N}(\mathbf{w},\Sigma)[\mathbf{u}]} d\pi_t(\theta) = \mathbb{E}_{P_t}[\mathbf{w}] = \mathbf{w}_t$.

One the other hand, due to the definition of the surrogate loss, we have $\exp\left(-\gamma \widetilde{f}_t(\mathbf{w}_t)/2\right) = 1$. We complete the proof by plugging the equality into the above displayed inequality. $\qquad \square$

The following lemma is a counterpart to Lemma 1 that incorporates the projection step. It shows that a similar mixability regret bound as Lemma 1 for the surrogate loss still holds.

**Lemma 8.** *Let $Q_t = \mathcal{N}(\mathbf{u}_t, \sigma^2 I_d)$ be a Gaussian distribution with mean $\mathbf{u}_t \in \mathcal{W}$ and covariance $\sigma^2 I_d \in \mathbb{R}^{d \times d}$ and define*

$$
\widetilde{m}_t(P) = -\frac{2}{\gamma} \ln \left( \mathbb{E}_{\mathbf{w} \sim P_t} \left[ \exp \left( -\frac{\gamma}{2} \widetilde{f}_t(\mathbf{w}) \right) \right] \right).
$$

*Algorithm 3 with the update rule (15), (16) and (17) with $\mu = 1/T$ ensures that the mixability regret is bounded as*

$$
\sum_{t=1}^T \widetilde{m}_t(P_t) - \sum_{t=1}^T \mathbb{E}_{\mathbf{u} \sim Q_t}[\widetilde{f}_t(\mathbf{u})] \leq \frac{4 + 2\mathrm{KL}(Q_1 \| P_1)}{\gamma} + \frac{2}{\gamma} \sum_{t=2}^T \int_{\mathbf{u} \in \mathbb{R}^d} (Q_t(\mathbf{u}) - Q_{t-1}(\mathbf{u})) \ln \left( \frac{1}{P_t(\mathbf{u})} \right) d\mathbf{u},
$$

*where $\mathrm{KL}(P\|Q) = \mathbb{E}_{\mathbf{u} \sim P}[\ln(P(\mathbf{u})/Q(\mathbf{u}))]$ refers to the Kullback-Leibler divergence.*

*Proof of Lemma 8.* Following the same reasoning as in the proof of Lemma 1, according to the exponential-weight update rule (15), we obtain

$$
\begin{aligned}
& \sum_{t=1}^T \widetilde{m}_t(P_t) - \sum_{t=1}^T \mathbb{E}_{\mathbf{u} \sim Q_t}[\widetilde{f}_t(\mathbf{u})] \\
&= \frac{2}{\gamma} \sum_{t=1}^T \left( \mathrm{KL}(Q_t \| P_t) - \mathrm{KL}(Q_t \| P'_{t+1}) \right) \\
&\leq \frac{2}{\gamma} \sum_{t=1}^T \left( \mathrm{KL}(Q_t \| P_t) - \mathrm{KL}(Q_t \| \widetilde{P}_{t+1}) \right) \\
&\leq \frac{2}{\gamma} \mathrm{KL}(Q_1 \| P_1) + \frac{2}{\gamma} \sum_{t=2}^T \left( \mathrm{KL}(Q_t \| P_t) - \mathrm{KL}(Q_{t-1} \| \widetilde{P}_t) \right) \\
&= \frac{2}{\gamma} \mathrm{KL}(Q_1 \| P_1) + \frac{2}{\gamma} \sum_{t=2}^T \int_{\mathbf{u} \in \mathbb{R}^d} (Q_t(\mathbf{u}) - Q_{t-1}(\mathbf{u})) \ln \left( \frac{1}{P_t(\mathbf{u})} \right) d\mathbf{u} + \frac{2}{\gamma} \sum_{t=1}^T \mathbb{E}_{\mathbf{u} \sim Q_{t-1}} \left[ \ln \left( \frac{\widetilde{P}_t(\mathbf{u})}{P_t(\mathbf{u})} \right) \right] \\
&\leq \frac{2}{\gamma} (2 + \mathrm{KL}(Q_1 \| P_1)) + \frac{2}{\gamma} \sum_{t=2}^T \int_{\mathbf{u} \in \mathbb{R}^d} (Q_t(\mathbf{u}) - Q_{t-1}(\mathbf{u})) \ln \left( \frac{1}{P_t(\mathbf{u})} \right) d\mathbf{u}
\end{aligned}
$$

where the first inequality follows from the generalized Pythagorean theorem for KL divergence. Specifically, since the set $\mathscr{M}$ is convex and closed in the topology of total variation distance as shown in Lemma 5, the projection $\widetilde{P}_{t+1} = \arg\min_{P \in \mathscr{M}} \mathrm{KL}(P \| P'_{t+1})$ exists according to Csiszár (1975, Theorem 2.1). Furthermore, Csiszár & Matus (2003) shows that that for any $Q_t \in \mathscr{M}$, we have $\mathrm{KL}(Q_t \| P'_{t+1}) \geq \mathrm{KL}(Q_t \| \widetilde{P}_{t+1})$. The last second equality follow the same reasoning in obtaining (27). The last inequality is due to the fixed-share step (17) with $\mu = 1/T$, where one can show $\mathbb{E}_{\mathbf{u} \sim Q_{t-1}} \left[ \ln \left( \widetilde{P}_t(\mathbf{u}) / P_t(\mathbf{u}) \right) \right] \leq 2/T$ by the same arguments in obtaining (28). $\qquad\square$

### C.3. Proof of Theorem 3

*Proof of Theorem 3.* For $\eta$-exp-concave loss function, Lemma 11 in Appendix D shows that the dynamic regret in terms of the original loss can be upper bounded by that in terms of the surrogate loss as

$$
\begin{aligned}
\text{D-REG}_T &\leq \sum_{t=1}^{T} \widetilde{f}_t(\mathbf{w}_t) - \sum_{t=1}^{T} \widetilde{f}_t(\mathbf{u}_t) \\
&= \underbrace{\sum_{t=1}^{T} \widetilde{f}_t(\mathbf{w}_t) - \widetilde{m}_t(P_t)}_{\text{TERM (A)}} + \underbrace{\sum_{t=1}^{T} \widetilde{m}_t(P_t) - \sum_{t=1}^{T} \mathbb{E}_{\mathbf{u} \sim Q_t}[\widetilde{f}_t(\mathbf{u})]}_{\text{TERM (B)}} + \underbrace{\sum_{t=1}^{T} \mathbb{E}_{\mathbf{u} \sim Q_t}[\widetilde{f}_t(\mathbf{u})] - \sum_{t=1}^{T} \widetilde{f}_t(\mathbf{u}_t)}_{\text{TERM (C)}},
\end{aligned}
$$

where $\widetilde{m}_t(P) = -\frac{2}{\gamma} \ln \left( \mathbb{E}_{\mathbf{w} \sim P_t} \left[ \exp \left( -\frac{\gamma}{2} \widetilde{f}_t(\mathbf{w}) \right) \right] \right)$ is the mix loss defined in terms of the surrogate loss.

①**Bounding mixability gap.** For TERM (A), Lemma 7 shows the mixability gap is non-positive as TERM (A) $\leq 0$.

②**Bounding mixability regret.** As for TERM (B), we have

$$
\begin{aligned}
\text{TERM (B)} &\leq \frac{4 + 2\mathrm{KL}(Q_1 \| P_1)}{\gamma} + \frac{2}{\gamma} \sum_{t=2}^{T} \int_{\mathbf{u} \in \mathbb{R}^d} (Q_t(\mathbf{u}) - Q_{t-1}(\mathbf{u})) \ln \left( \frac{1}{P_t(\mathbf{u})} \right) \mathrm{d}\mathbf{u} \\
&\leq \frac{2}{\gamma} \left( 2 + d\ln\left(\frac{1}{\sigma}\right) + \frac{d\sigma^2}{2} + \frac{D^2}{2} \right) + \frac{2}{\gamma} \sum_{t=2}^{T} \int_{\mathbf{u} \in \mathbb{R}^d} (Q_t(\mathbf{u}) - Q_{t-1}(\mathbf{u})) \ln \left( \frac{1}{P_t(\mathbf{u})} \right) \mathrm{d}\mathbf{u} \quad (48)
\end{aligned}
$$

where the first inequality is due to Lemma 8 in Appendix C.2, which serves as the counterpart to Lemma 1 when the projection step is incorporated. The second inequality follows from the definitions $Q_1 = \mathcal{N}(\mathbf{u}_1, \sigma^2 I_d)$ and $P_1 = \mathcal{N}(\mathbf{w}_0, I_d)$, together with Lemma 10.

For the last term on the right-hand side of (48), we follow the same argument as in the proof of Theorem 1 and decompose the integral into three regions: $\mathcal{W}_t^{(1)} = \{\mathbf{u} \in \mathbb{R}^d \mid Q_t(\mathbf{u}) < Q_{t-1}(\mathbf{u}) \text{ and } P_t(\mathbf{u}) > 1\}$, $\mathcal{W}_t^{(2)} = \{\mathbf{u} \in \mathbb{R}^d \mid Q_t(\mathbf{u}) > Q_{t-1}(\mathbf{u}) \text{ and } P_t(\mathbf{u}) < 1\}$, and the remainder $\overline{\mathcal{W}}_t = \mathbb{R}^d \setminus \mathcal{W}_t^{(1)} \cup \mathcal{W}_t^{(2)}$. Since $(Q_t(\mathbf{u}) - Q_{t-1}(\mathbf{u})) \ln(1/P_t(\mathbf{u})) \leq 0$ for all $\mathbf{u} \in \overline{\mathcal{W}}_t$, the integral over this region is non-positive. Therefore, it suffices to bound the integrals over $\mathcal{W}_t^{(1)}$ and $\mathcal{W}_t^{(2)}$. For the integral over region $\mathcal{W}_t^{(1)}$, We have

$$
\int_{\mathbf{u} \in \mathcal{W}_t^{(1)}} (Q_t(\mathbf{u}) - Q_{t-1}(\mathbf{u})) \ln \left( \frac{1}{P_t(\mathbf{u})} \right) \mathrm{d}\mathbf{u} \leq \frac{d}{2} \ln \left( \frac{T}{2\pi} \right) \cdot \|Q_t - Q_{t-1}\|_1, \quad (49)
$$

where (49) holds by $P_t = \mu N_0 + (1 - \mu)\widetilde{P}_t \in \mathscr{M}$ and $\max_{\mathbf{u} \in \mathbb{R}^d} \ln(P_t(\mathbf{u})) \leq \frac{d}{2} \ln\left(\frac{T}{2\pi}\right)$ as shown in Lemma 6.

As for the integral over the region $\mathcal{W}_t^{(2)}$, due to the fixed-share update rule (17), the same arguments in obtaining (33) in the proof of Theorem 1 can be applied to show that

$$
\int_{\mathbf{u} \in \mathcal{W}_t^{(2)}} (Q_t(\mathbf{u}) - Q_{t-1}(\mathbf{u})) \ln \left( \frac{1}{P_t(\mathbf{u})} \right) \mathrm{d}\mathbf{u} \leq \left( \frac{d\ln(2\pi)}{2} + D + \ln\left(\frac{1}{\mu}\right) \right) \|Q_t - Q_{t-1}\|_1 + 2d\sigma^2, \quad (50)
$$

Then, plugging (49) and (50) back into (48) and further upper bound the total variation between $Q_t$ and $Q_t$ by the path-length as shown in (35), we have

$$\text{TERM (B)} \leq \frac{(d+2)\ln T + D}{\gamma} \cdot \frac{P_T}{\sigma} + \frac{d\sigma^2(4T+1)}{\gamma} + \frac{2D^2+4}{\gamma} + \frac{2d}{\gamma}\ln\left(\frac{1}{\sigma}\right). \tag{51}$$

③ **Bounding comparator gap.** As for TERM (C), given the definition of the loss function $\widetilde{f}_t$, the comparator gap can be directly calculated by

$$
\begin{aligned}
\text{TERM (C)} &= \sum_{t=1}^{T} \mathbb{E}_{\mathbf{u}\sim Q_t}[\widetilde{f}_t(\mathbf{u}) - \widetilde{f}_t(\mathbf{u}_t)] \\
&= \sum_{t=1}^{T} \mathbb{E}_{\mathbf{u}\sim Q_t}\left[\mathbf{g}_t^\top(\mathbf{u}-\mathbf{u}_t)\right] + \frac{\gamma}{2}\mathbb{E}_{\mathbf{u}\sim Q_t}\left[\left(\mathbf{g}_t^\top(\mathbf{u}-\mathbf{w}_t)\right)^2 - \left(\mathbf{g}_t^\top(\mathbf{u}_t-\mathbf{w}_t)\right)^2\right] \\
&= \sum_{t=1}^{T} \frac{\gamma}{2}\mathbb{E}_{\mathbf{u}\sim Q_t}[(\mathbf{g}_t^\top(\mathbf{u}-\mathbf{u}_t))^2] \\
&\leq \sum_{t=1}^{T} \frac{\gamma\|\mathbf{g}_t\|_2^2}{2} \cdot \mathbb{E}_{\mathbf{u}\sim Q_t}[\|\mathbf{u}-\mathbf{u}_t\|_2^2] \\
&\leq \frac{\gamma d\sigma^2 \sum_{t=1}^{T}\|\mathbf{g}_t\|_2^2}{2} \leq \frac{\gamma d\sigma^2 G^2 T}{2}
\end{aligned}
$$

where the third equality is due to $\mathbb{E}_{\mathbf{u}\sim Q_t}[\mathbf{u}] = \mathbf{u}_t$. The last second inequality is by $\mathbb{E}_{\mathbf{u}\sim Q_t}[\|\mathbf{u}-\mathbf{u}_t\|_2^2] \leq (d\sigma^2)/2$.

Then, combining the upper bound for TERM (A), TERM (B) and TERM (C), we have

$$\text{D-REG}_T \leq \frac{(d+2)\ln T + D}{\gamma} \cdot \frac{P_T}{\sigma} + d\left(\frac{5}{\gamma} + \frac{\gamma G^2}{2}\right)\sigma^2 T + \frac{2D^2+4}{\gamma} + \frac{2d}{\gamma}\ln\left(\frac{1}{\sigma}\right).$$

We can further bound the above regret bound by considering different value of $P_T$.

**Case 1:** $P_T \leq \sqrt{\frac{1}{T}}$. We can choose $\sigma = \sqrt{1/T}$ to obtain $\text{D-REG}_T \leq \mathcal{O}(d\log T)$.

**Case 2:** $P_T \geq \sqrt{\frac{1}{T}}$. We select $\sigma = P_T^{\frac{1}{3}} T^{-\frac{1}{3}}$ to obtain $\text{D-REG}_T \leq \mathcal{O}\left(d\log T \cdot T^{\frac{1}{3}} P_T^{\frac{2}{3}} + d\log T\right)$. □

## D. Technical Lemmas

In this section, we provide several useful lemmas used in the proof.

**Lemma 9** (Theorem 1.8.1 of Ihara (1993)). *Let $Q = \mathcal{N}(\mathbf{u}_q, \Sigma_q)$ be a d-dimensional Gaussian distribution. Then the entropy of $Q$ is given by*

$$H(Q) = \frac{d}{2}\ln(2\pi e) + \frac{1}{2}\ln|\Sigma_q|.$$

**Lemma 10** (Theorem 1.8.2 of Ihara (1993)). *The Kullback-Leibler divergence between two d-dimensional Gaussian distributions $P = \mathcal{N}(\mathbf{u}_p, \Sigma_P)$ and $Q = \mathcal{N}(\mathbf{u}_q, \Sigma_q)$ is given by*

$$\text{KL}(Q\|P) = \frac{1}{2}\left(\ln\left(\frac{|\Sigma_p|}{|\Sigma_q|}\right) + \text{Tr}(\Sigma_q\Sigma_p^{-1}) + \|\mathbf{u}_p-\mathbf{u}_q\|_{\Sigma_p^{-1}}^2 - d\right).$$

**Lemma 11** (Lemma 4.2 of Hazan (2016)). *Let $\mathcal{W} \subseteq \mathbb{R}^d$ be a convex and closed set. Suppose $\|\mathbf{u}-\mathbf{v}\|_2 \leq D$ holds for any $\mathbf{u}, \mathbf{v} \in \mathcal{W}$ and $\|\nabla f(\mathbf{w})\|_2 \leq G$ holds for any $\mathbf{w} \in \mathcal{W}$. For any $\eta$-exp-concave loss, the following holds for all $\gamma \leq \min\{1/(8GD), \eta/2\}$ and $\mathbf{w}, \mathbf{u} \in \mathcal{W}$:*

$$f(\mathbf{w}) - f(\mathbf{u}) \leq \nabla f(\mathbf{w})^\top(\mathbf{w}-\mathbf{u}) + \gamma\left(\nabla f(\mathbf{w})^\top(\mathbf{w}-\mathbf{u})\right)^2.$$

**Lemma 12** (Lemma 10 of van Erven & Koolen (2016)). *Let $0 < \eta \le \frac{1}{5GD}$. Consider a Gaussian distribution with mean $\boldsymbol{\mu} \in \mathcal{W}$ and arbitrary symmetric positive-definite covariance $\Sigma$, where $\mathcal{W} \subseteq \mathbb{R}^d$ is a convex and compact set. Let $\ell_t^\eta(\mathbf{w}) = -\eta\langle \mathbf{g}_t, \mathbf{w}_t - \mathbf{w}\rangle + \eta^2(\mathbf{w}_t - \mathbf{w})^\top \mathbf{g}_t \mathbf{g}_t^\top (\mathbf{w}_t - \mathbf{w})$. Then,*

$$\mathbb{E}_{\mathbf{w} \sim \mathcal{N}(\boldsymbol{\mu}, \Sigma)}\big[ \exp(-\ell_t^\eta(\mathbf{w})) \big] \le \exp(-\ell_t^\eta(\boldsymbol{\mu})),$$

*where the domain $\mathcal{W}$ is bounded by $D$ such that $\|\mathbf{w} - \mathbf{w}'\|_2 \le D$ for all $\mathbf{w} \in \mathcal{W}$ and $\|\mathbf{g}_t\|_2 \le G$.*

