# OpenReview forum: "Non-stationary Online Learning for Curved Losses: Improved Dynamic Regret via Mixability"
_ICML.cc/2025/Conference — ICML 2025 poster_

### Official Review · Reviewer_Ukmi · 2025-03-12

**Overall Recommendation:** 2

**Summary:**

This paper shows that the fixed share algorithm is able to obtain optimal dynamic regret under mixable losses with improper online learning. They also obtain the first gradient variation based dynamic regret bounds under curved losses. The results are novel to the best of my knowledge.

### Post Rebuttal ####
I have read the authors' responses and comments by other reviewers. I think this paper definitely makes a significant progress in dynamic regret literature. I also agree with other reviewers that the presentation of the paper can be greatly improved along with a more coherent discussion regarding prior works.

The comment from authors regarding the applicability of improper to proper reduction under linear / logistic regression is not convincing enough. Note that in this case the availability of covariate before making the prediction allows us to know the gradient of the loss upto a scaling factor. This information along with exp-concavity of losses can be exploited for constructing the reduction.

I believe that a careful revision that addresses the above concerns can greatly benefit the readers and can make the paper more complete and impactful. Based on this, I would like to maintain my score.

**Claims And Evidence:**

Yes

**Essential References Not Discussed:**

The authors seem unaware about the following important references. In Table 2, the problems of optimal dynamic regret for proper learning under linear and logistic regression losses were reported to be unsolved. However, they have been addressed affirmatively in the following two papers.

1) Optimal Dynamic Regret in LQR Control, D Baby and YX Wang, NeurIPS 2022 (solves multi-task linear regression)

2) Non-stationary Contextual Pricing with Safety Constraints, D Baby, J Xu and YX Wang, TMLR 2023 (solves general GLM type losses, including logistic regression)

**Experimental Designs Or Analyses:**

Yes.

**Methods And Evaluation Criteria:**

Yes

**Other Comments Or Suggestions:**

See Questions For Authors.

**Other Strengths And Weaknesses:**

See Questions For Authors.

**Questions For Authors:**

I believe this paper provides novel insights into a hard problem. My only complaint is that I would like to see a more careful comparison with prior works of Baby and Wang on this topic. This helps to place the current work appropriately in the literature and provides researchers to better understand the pros and cons of each approach.

1) The current work requires to know the form of losses beforehand to compute the output of mixability mapping. Baby and Wang 2021, 2022 do not require to know the form of losses beforehand. Specifically Baby and Wang 2022, requires only access to gradients of the losses.

2) The current work assumes gradient Lipschitnzess for the losses while bounding the comparator gap. I did not see this explicitly stated in any assumptions, but only mentioned in the proof sketch. The work of Baby and Wang 2022 does not require this assumption.

3) Results of Baby and Wang 2021, 2022 can also work with sleeping experts style experts. This reduces the per-round complexity logarithmically. For example, for squared error loss, we obtain a per-round complexity of $O(\log T)$ while blowing the regret up by a logarithmic factor. Does something similar hold true for your approach?

4) In Table 2, I would like to see a comparison regarding run-times as well as the inclusion of prior works that solves the problem of proper learning under linear and logistic regression losses mentioned in the section Essential References Not Discussed above. This way the comparison will be more complete and fair and will directly convey the readers about the merits and demerits of each approach.

5) Are there examples of losses that are mixable but not exp-concave in a compact domain?

I would be happy to see this manuscript accepted in a venue after including the aforementioned details.

**Relation To Broader Scientific Literature:**

Discussed in the related work of the paper.

**Theoretical Claims:**

Math was not checked in detail.

---

> ### Author Rebuttal · Authors · 2025-04-01
>
> Thanks for your expert comments! We will address your main concerns regarding the literature comparisons. Without a doubt, Baby and Wang pioneered the line of dynamic regret for exp-concave/strongly convex functions. While we attempted to make a comparison, we unfortunately missed two relevant references, which will be included in the revised version. Moreover, we will add a paragraph discussing the limitations of fixed-share-type methods, which is attached at the end of this rebuttal.
>
> ---
>
> **Q1:** "requires to know the form of losses beforehand"
>
> **A1:** Thank you for pointing this out. Yes, constructing the mixability mapping requires focusing on specific loss forms. While this may be a limitation of mixability-based methods, the form of loss is often predetermined in many applications. Examples include online nonparametric regression, online classification, and LQR control, as the reviewer noted. We will clarify this in the next revision
>
> ---
> **Q2:** "current work assumes gradient Lipschitnzess for the losses"
>
> **A2:** Thank you for the comments. In Theorem 1, we state that $\beta$-smoothness is required to achieve the improved bound. We believe this is a mild assumption, as many curved losses, like those in Section 4, are smooth. That said, removing the smoothness assumption, as done by Baby and Wang et al. (2022), remains a quite interesting question. We will clarify this comparison in the next revision.
>
> ---
>
> **Q3:** "Results of Baby and Wang 2021, 2022 can also work with sleeping experts style experts."
>
> **A3**: Currently, achieving $O(\log T)$ time complexity for the fixed-share method is challenging due to the need to add a small portion of the prior distribution $N_0$ at each iteration. We will clarify the time-complexity comparison in the revision.
>
> ---
>
> **Q4:** two missing references and comparisons
> > - unaware about the following important references...
> > - Table 2: comparison regarding run-times as well as the inclusion of prior works.
>
> **A4:** We will definitely include these two papers and update the table to include the time complexity of the compared methods.
>
> $\dagger$ Indicates that the time complexity can be improved to $O(\log T)$ using more refined geometric covering techniques.
>
> | Losses              | Method                                | Regret Bound                                    | Proper Learning | Time Complexity  |
> | - | -| - |- |- |
> | Least-squares loss  | Theorem 5 (Baby and Wang, NeurIPS'22) | $\widetilde{O}(d + d^{10/3} T^{1/3} P_T^{2/3})$ | Yes            | $O(T)^{\dagger}$ |
> | Logistic regression | Theorem 3.1 (Baby et al., TMLR'23)    | $\widetilde{O}(d + d^{10/3} T^{1/3} P_T^{2/3})$ | Yes            | $O(T)^{\dagger}$ |
>
> ---
>
> **Q5:** losses that are mixable but not exp-concave in a compact domain?
>
> **A5:** To our knowledge, common mixable losses, like squared loss, logistic loss are also exp-concave with generally larger coefficients. A key difference is that the mixability coefficient $\eta_{\mathtt{mix}}$ typically does not depend on the diameter of the decision domain $\mathcal{W}$, whereas the exp-concavity coefficient $\eta_{\mathtt{exp}}$ often does. For example, the logistic loss is 1-mixable over $\mathbb{R}^d$ but only $e^{-D}$-exp-concave within a bounded domain $\mathcal{W}$ of diameter $D$. Consequently, mixability-based results may extend to unconstrained comparators, while exp-concavity-based methods generally require bounded domains.
>
> ---
> [Revision: Section 4.4]
> **Limitation and Future Work:**  Although our method achieves improved dynamic regret without relying on KKT-based analysis, there remain several directions for improvement.  First, regarding time complexity: our method matches the $O(t)$ complexity of Baby and Wang et al. (2021, 2022) for the squared loss. However, prior work can further reduces this to $O(\log T)$ by using the sleeping expert algorithm with a geometric cover, incurring additional multiplicative logarithmic factors in regret. Incorporating this idea into the fixed-share update is a promising direction.  Second, while common curved losses like squared loss and logistic regression are smooth, the analysis by Baby and Wang et al. (2022) does not require smoothness. Removing the smoothness assumption in mixability-based analysis remains an interesting theoretical challenge.  Finally, to construct the mixable prediction $z_{\mathtt{mix}}$, mixability-based methods require the loss function to have a specific form, unlike the approach by Baby and Wang et al. Although this condition holds in many applications—such as online nonparametric regression, online logistic regression, and LQR control—it is worth exploring whether we can aggregate distributions efficiently using exp-concavity instead of mixability.
>
> ---
>
> We will add these discussions and include a fair comparison to Baby and Wang's line of work. Please consider updating the score if our responses and the additions have adequately addressed your concerns. Thank you!

---

> > ### Comment · Reviewer_Ukmi · 2025-04-08
> >
> > Thank you for the response. I have one more question towards the authors in regards to extension of their results to the setting of proper learning. For the case of linear / logistic regression where the covariate is revealed before making a prediction, could you please comment on what the main blocker is when using the improper-proper online learner reduction that is constructed in Baby, Xu and Wang 2023 (https://openreview.net/pdf?id=fWIQ9Oaao0)?

---

> > > ### Author Response · Authors · 2025-04-09
> > >
> > > Thank you for this insightful question. It appears challenging to apply the technique from Baby et al. (2023) to our method, mainly because the source of improper learning differs. In the earlier work (Baby et al., 2021), improper learning is necessary because the method requires learning over an extended box domain. In such a case, the idea of Cutkosky & Orabona (2018) can be used to ensure proper learning by carefully designing the surrogate loss outside of the decision domain.
> > >
> > > In contrast, the source of improper learning of our method arises from the aggregation step. As shown in Equations (13) and (15), our prediction could be non-linear. This type of non-linear predictor resembles the one used in the VAW method, and it may be difficult to apply the technique from Baby et al. (2023) to address this issue.
> > >
> > > One potential direction is to aggregate distributions using exp-concavity rather than mixability, which may allow for a linear predictor. However, a notable challenge with this approach is that the exp-concavity coefficient may be uncontrolled.
> > >
> > > Although proper learning can be achieved in certain interesting cases, the distinction between proper and improper learning remains a key mystery in the study of dynamic regret for curved functions. Exploring methods to achieve proper learning within our framework is an important direction for future work.

---

### Official Review · Reviewer_NuLh · 2025-03-13

**Overall Recommendation:** 3

**Summary:**

This paper studies non-stationary online convex optimization with mixable loss functions. The class of mixable function includes the exp-concave functions. This paper proposes a fixed-share algorithm for continuous space. In each round, the proposed algorithm requires to obtain a decision satisfying a certain inequality concerning loss function in the round. The paper demonstrates that such a decision can be obtained when the loss function is one of squared, least-squares, or logistic losses. The proposed algorithm achieves $O(d\log T + (d + \log(T / P_T)) T^{1/3}P_T^{2/3})$-dynamic regret, significantly improving existing results with respect to $d$. Furthermore, it offers a simpler and more comprehensible proof compared to previous analyses.

## update after rebuttal
Thank you for your response.
My concerns have been addressed, so I raised my score.

I would like to draw your attention to the fact that the definition of mixable functions will change, so the logarithmic function (example 3) will no longer be an appropriate example.

**Claims And Evidence:**

The paper claims to utilize the concept of mixability for analysis. However, it actually relies on stronger assumptions. Specifically, the proposed algorithm requires $w \in \mathcal{W}$ that satisfies equation (4) for the loss function, which differs slightly from the inequality in the definition of a mixable function (Definition 2). Since Definition 2 deals with distributions whose support is $\mathcal{W}$ and equation (4) handles distributions supported on $\mathbb{R}^d$, it has not been shown that mixable functions always possess $w$ that satisfies equation (4). Moreover, Theorem 1 supposes that the diameter of $\mathcal{W}$ is at most $D$, ensuring these inequalities do not align. Thus, as I understand it, the paper employs mixability-inspired conditions for its analysis rather than direct mixability.

**Essential References Not Discussed:**

It appears the necessary prior works have been appropriately cited and discussed to comprehend the contributions.

**Experimental Designs Or Analyses:**

There don't appear to be discrepancies in the design or methodology, and they are clearly explained.

**Methods And Evaluation Criteria:**

The usage of dynamic regret as a performance metric for non-stationary online learning seems appropriate. The analyses indicate that the proposed algorithm achieves near-optimal performance.

**Other Comments Or Suggestions:**

- line 78 (right column): meethod -> method
- Eq. (6): N_0(u) -> N_0
- There is inconsistency in the order of arguments in mixed losses. In squared loss, it's denoted as $m(P, y)$, but in least-square loss as $m(y, P)$.
- Proof of Corollary 1
  - lines 944-945: inequality (3) -> inequality (4)
  - line 945: (3) is essentially -> (13) is essentially
  - line 956: give a fixed $z$ -> given a fixed $z$
- Appendix B.3 should be integrated into Appendix A.

**Other Strengths And Weaknesses:**

Though the algorithm's optimal performance is restricted to some typical loss functions, the novel ideas and straightforward analysis constitutes a noteworthy contribution. The paper is generally well-written and accessible. However, the paper's connection to mixable functions deserves a more precise discussion.

**Questions For Authors:**

1. Could you provide a more detailed proof regarding why equation (13) satisfies equation (4)? Specifically, it is not obvious to me that equation (13) is the optimal solution of the optimization problem defined on line 959.

**Relation To Broader Scientific Literature:**

While the paper's results are limited to typical exp-concave functions, it achieves near-optimal regret bounds through highly versatile analysis using the conditions inspired by mixability.

**Theoretical Claims:**

I examined the proof sketch for Theorem 1, the proofs of Theorem 3, and Corollaries 1, 2, and 3. In Corollary 1, I am unclear on why $z$, as defined in equation (13), satisfies equation (4), potentially due to omitted critical arguments.

---

> ### Author Rebuttal · Authors · 2025-04-01
>
> Thanks for your very careful review and pointing out two technical problems. We have provided a detailed proof for equation (13), and clarify that our Theorem 1 indeed requires the mixability of loss functions over $\mathbb{R}^d$. These issues will not affect the key contributions of our paper, but we acknowledge that they were not stated with sufficient precision. We will carefully revise the paper to address these imprecisions and ensure clarity.
>
> ---
>
> **Q1:** the optimal solution of line 959.
>
> **A1:** Thanks for the detailed inspection of our proof. Here, we provide a more thorough explanation of why equation (13) represents the optimal solution.
>
> *Proof.* First, observe that $m_{sq}(P, y)$ is constant for a given value of $y$. Let us define $M_1 = m_{sq}(P, B)$ and $M_2 = m_{sq}(P, -B)$. The optimization problem in line 959 can then be reformulated as:
>
> $~~~~~~~\begin{equation}\arg\min_{z \in \mathbb{R}^d} \max\\{(z - B)^2 - M_1, (z + B)^2 - M_2\\}.\end{equation}$
>
> Define the functions $g_1(z) = (z - B)^2 - M_1$ and $g_2(z) = (z + B)^2 - M_2$. We have the follow close-form solution for the inner optimzation problem:
>
> $~~~~~~~\begin{equation} h(z) = \max\\{g_1(z), g_2(z)\\} =  \begin{cases} g_1(z), \& \text{if } z \leq z_*, \\\ g_2(z), \& \text{otherwise}, \end{cases}\end{equation}$
>
> where $z_* = \frac{M_2 - M_1}{4B}$. We now analyze three separate cases:
>
> 1. When $-B \leq z_* \leq B$:
>
>    - For $z \in (-\infty, z_*]$, $h(z) = g_1(z)$, which is decreasing since $z_* \leq B$.
>    - For $z \in [z_*, \infty)$, $h(z) = g_2(z)$, which is increasing since $z_* \geq -B$.
>
>    Hence, the minimum of $h(z)$ is achieved at $z = z_*$.
>
> 2. When $z_* < -B$:
>
>    - On $(-\infty, z_*]$, $h(z) = g_1(z)$ remains decreasing, and the minimum in this interval is at $z = z_*$.
>    - On $(z_*, \infty)$, the minimum of $h(z)$ is at $z = -B$ since $h(-B) = g_2(-B) = -M_2$.
>
>    Under the condition $z_* \leq -B$, one can verify that $h(-B) \leq h(z_*)$, so the overall minimum is attained at $z = -B$.
>
> 3. When $z_* > B$: By symmetry to the previous case, the minimum of $h(z)$ occurs at $z = B$ using the same reasoning.
>
> By combining all three cases, we can show that the optimal solution is given in equation (13). We will clarify this reasoning in the revised version of the manuscript. $\blacksquare$
>
> ---
>
> **Q2:** ".... it actually relies on stronger assumptions, ...Definition 2..., ...Theorem 1..." ($\mathcal{W}$ versus $\mathbb{R}^d$).
>
> **A2:** Thanks for pointing out this issue. Theorem 1 indeed requires the mixability of loss functions over $\mathbb{R}^d$. We apologize for the confusion caused by the imprecise statements. We now describe the assumptions and decision domain more precisely.
>
> - **On assumption:** Our assumption for Algorithm 1 is that the loss function is mixable over $\mathbb{R}^d$, which is necessary due to the use of a Gaussian prior in the algorithm. This assumption is satisfied by several commonly used curved loss functions in online learning, such as the squared loss and logistic regression loss, as discussed in Examples 1 and 2 and elaborated on in Section 4. Moreover, while the algorithm maintains a distribution over $\mathbb{R}^d$, we emphasize that for certain losses—such as the squared loss—the resulting final predictor $w$ can still lie within the domain $\mathcal{W}$, as shown in Section 4.1.
> - **On decision domain:** In the generic algorithmic template, we allow the improper learning where the predictor $\mathbf{w}_t$ is outside the domain $\mathcal{W}$, while the comparator remains constrained within $\mathcal{W}$. Therefore, the proposed method may be improper depending on the loss function. For the squared loss, we obtain a *proper* learning algorithm, but for least-squares regression and logistic regression, the method should be considered *improper*, as discussed in Section 4.
>
> ---
>
> To ensure that the assumptions and problem setting are clearly presented for the generic template, we will add a new subsection at the beginning of Section 3 to explicitly state the assumptions and capabilities of the learner. We will add the proofs for equation (13) in the paper.
>
> Please consider updating the score if our responses and the additions in the revised version have adequately addressed your concerns. Thank you!

---

### Official Review · Reviewer_jqG3 · 2025-03-14

**Overall Recommendation:** 3

**Summary:**

This work proposes an algorithm for non-stationary online learning under mixable losses. They provide better dynamic regret bounds in comparison to the existing results in terms of the dependence on the dimension and logarithmic redundancy.

## update after rebuttal
I keep my score which remains positive.

**Claims And Evidence:**

I do not see any problematic claims.

**Essential References Not Discussed:**

There does not appear to be missing essential references.

**Experimental Designs Or Analyses:**

No numerical experiments.

**Methods And Evaluation Criteria:**

I do not see any major issues with the methods.

**Other Comments Or Suggestions:**

The paper will benefit from an earlier detailed comparison. I suggest the authors to provide a detailed comparison table in page 2 and move section 4.4 to section 1 as well. The comparison table should extend Table 1, still including the prior works, especially Baby & Wang, 2021; 2022. In addition to specific settings, compare their general regret results as well (possibly using the mixability coefficient). It should also provide differing assumptions if any.

Also, correct the typos such as "mixablity" in lines 204 and 208.

**Other Strengths And Weaknesses:**

While the dynamic regret improvement is a strength, comparisons to the existing literature are a bit lacking. While some comparisons to Baby & Wang, 2021; 2022 are made, they should be discussed in more detail. The proper learning setting should be more clearly discussed and compared. When comparing, altering the regret dependency on dimension $d$ by relying on $L_1$ and $L_2$ norm relation should be more adequately discussed and motivated.

The results appear new and it seems the main idea is to exploit the mixability property as opposed to utilizing the primal and dual variable structure imposed by the KKT conditions. The manuscript would benefit from a more detailed comparison of the proof techniques to further legitimize your claims.

**Questions For Authors:**

What different techniques do you use in comparison to Baby & Wang (2021; 2022) to achieve the improved dynamic regret bounds?

**Relation To Broader Scientific Literature:**

Improved dynamic regret bounds for mixable or exp-concave losses are valuable for the online learning literature.

**Theoretical Claims:**

The proofs seem correct.

---

> ### Author Rebuttal · Authors · 2025-04-01
>
> We thank the reviewer for appreciating the novelty of our methods and for the constructive suggestions. In the revision, we will include a more detailed comparison with Baby and Wang (2021) in the introduction, highlighting the issue of proper learning and the underlying assumptions. Below, we address the questions regarding the differences in proof techniques.
>
> **Q1:** The manuscript would benefit from a more detailed comparison of the proof techniques to further legitimize your claims.
>
> **A1**: Thank you for the helpful comments. Compared with Baby and Wang (2021) and subsequent literature, our work adopts a completely different analytic framework to achieve an improved dynamic regret bound. This distinction is evident in both the algorithmic design and the theoretical analysis.
>
> - *From the perspective of algorithmic design*, the main challenge is how to adapt to non-stationary environments without prior knowledge of $V_T$. Baby and Wang (2021) and the subsequent work address this by employing a strongly adaptive algorithm to ensure adaptivity. In contrast, our method leverages the idea of fixed-share updates over a continuous space. This fundamental difference in algorithmic approach leads to a completely different proof technique in the analysis.
>
> - *On the theoretical side*, the core challenge in Baby and Wang (2021) lies in proving improved dynamic regret for a strongly adaptive algorithm. This requires using KKT conditions to characterize an offline optimal sequence and demonstrating that a strongly adaptive algorithm can track this sequence with low cost. In contrast, our analysis is based on the concept of mixability. By carefully designing the comparator distribution $Q_t$, we are able to directly establish the optimal improved dynamic regret bound.
>
> We will clarify these points in the revision at the introduction level.

---

### Official Review · Reviewer_xQcX · 2025-03-17

**Overall Recommendation:** 2

**Summary:**

This paper considers online convex optimization (OCO) with mixable stage cost functions. The paper proposesseveral algorithms based on exponential weights with fixed share updates to achieve an improved dynamic regret bound than the bound in (Baby & Wang 2021). The improvements are in two aspects: improvement dependence on dimension d, and a slight improvement on the log(T) term.

**Claims And Evidence:**

This is a technical paper with no simulations. All the theorems and lemmas are clearly supported by proofs.

**Essential References Not Discussed:**

Since the mixability is closely related with strong convexity and one supporting example is quadratic loss which is strongly convex, the paper should also review the dynamic regret analysis for OCO with strongly convex costs, for example [1] discusses the fundamental lower bound of dynamic regret for OCO with strong convexity in Theorem 1.

[1] Li, Y., Qu, G. and Li, N., 2020. Online optimization with predictions and switching costs: Fast algorithms and the fundamental limit. IEEE Transactions on Automatic Control, 66(10), pp.4761-4768.

**Experimental Designs Or Analyses:**

There is no simulation provided.

**Methods And Evaluation Criteria:**

Yes, the proposed method is evaluated by dynamic regret, which is proper and widely used in OCO literature.

**Other Comments Or Suggestions:**

1. The paper should have a (sub)section devoted to problem formulation, with settings and assumptions in one place and stated explicitly. For example, do you need W to be bounded? Do you need f_t to have some smoothness properties?

2. It is better to provide some overview for the algorithms in all sections. The paper first introduced Algo1, which gives no explanation on implementation issues, and only discusses how to implement it after two sections. It is very confusing for the reader at their first read.

**Other Strengths And Weaknesses:**

Strengths:

1. The paper explores a different property in the convex functions that enjoy applications in logistic regression. Based on this property, this paper proposes novel algorithms to exploit this property and achieve better dynamic regret bounds.

2. The illustration Figure 1 is helpful for clarity.

3. The discussions on implementation of the proposed algorithms on three specific examples in Section 4 is also very helpful for understanding the paper and implementing the proposed algorithms.

Weaknesses

1. The paper is not very easy to read in general. There is no section for problem formulation, some assumptions and problem settings are in Section 1, some in Section 2, some in Section 3. It is very difficult to read and understand the key setting and the assumptions needed for the proposed algorithm and the regret analysis.

2. Though the paper discusses the implementation of the algorithms in three examples, it is still challenging to see how the algorithm can be implemented for general cost functions, especially the condition (4).

3. There is no simulation provided, casting doubt on the applicability of the proposed algorithms.

4. The paper could benefit from more discussions on the computation complexity of the proposed method, especially when the cost functions are in general form.

**Questions For Authors:**

See above.

**Relation To Broader Scientific Literature:**

This paper considers a different property, mixability,  on the OCO stage cost function than the commonly considered convexity and strong convexity. A new algorithm is proposed to leverage this different property.

**Theoretical Claims:**

The proofs seem correct after a quick read.

---

> ### Author Rebuttal · Authors · 2025-04-01
>
> We thank the reviewer for very helpful comments. The main concerns are about i) presentation and ii) technical questions on the construction of mixability prediction for general functions. We first provide a concise answer and will expand the details later.
>
> - **[On Presentation]** Our paper indeed requires much improvement, and we will enhance the presentation according to your constructive suggestions. Specifically, we will: include a subsection for problem formulation, reorganize the presentation of algorithms, and expand on the implementation of key conditions, etc. These changes aim to ensure the paper is easier to read.
> - **[On mixability prediction]** For several common functions of interests (like square loss for regression, logistic loss for classification), the mixability prediction can be explicitly and efficiently constructed. For other ones, there may not have a close-form and one can obtain a feasible prediction by solving min-max optimization problems.
>
> ---
>
> **Q1:** one paper presentation and assumptions
>
> > - "paper should have a (sub)section devoted to problem formulation"
> > - "...do you need W to be bounded? Do you need $f_t$ to have some smoothness properties?"
>
> **A1:** In the original submission, we stated the assumptions within the theorem statements.  For Theorem 1, it requires the loss functions to be $\eta$-mixable and $\beta$-smooth, along with the boundedness of the comparator domain. One potential confusion is that we allow *improper* predictions, that is,  the predictor $\mathbf{w}_t$ can be unconstrained or take a nonlinear form—such as equations (14) or (15), which are used for least squares and logistic regression.
>
> Thanks for your comments, we will revise the paper to consolidate all assumptions and the problem setting into a separate subsection at the beginning of Section 3 for improved clarify.
>
> ---
>
> **Q2:** about algorithm presentation
>
> > "first introduced Algo1, which gives no explanation on implementation issues, and only discusses how to implement it after two sections"
>
> **A2:** The goal of Section 3.1 is to present a generic algorithmic template along with conditions that guarantee improved dynamic regret. However, we agree that deferring the discussion of specific implementations may make the algorithm less accessible. In light of this, we plan to add the following remark in Section 3.1 to provide earlier insight into the implementation:
>
> **[Revision at Section 3.1]**: *Remark 1:* Condition (4) is equivalent to the mixability condition defined in Definition 2, which ensures that for any function that is mixable over $\mathbb{R}^d$, a predictor satisfying condition (4) always exists. In the context of online prediction with a loss function of the form $f_t(\mathbf{w}_t) = \ell(\mathbf{w}_t^\top\mathbf{x}_t, y_t)$, where $(\mathbf{x}_t, y_t)$ is the feature-label pair, Vovk (1999, Equations 11 and 12) and Cesa-Bianchi and Lugosi (2006, Proposition 3.3) provide a general optimization framework for constructing such predictors (which may be improper). Moreover, for the squared loss and logistic loss, closed-form constructions of the predictor are available; these are discussed further in Section 4.
>
> ---
>
> **Q3**: about the condition (4)
>
> > - ... it is still challenging to see how the algorithm can be implemented for general cost functions, especially the condition (4)...
> >
> > - the computation complexity of the proposed method, especially when the cost functions are in general form.
>
> A3: As mentioned in the response to *Q2*, for the online prediction problem where $f_t(\mathbf{w}) = \ell(\mathbf{w}^\top\mathbf{x}_t, y_t)$, Vovk (1999, Equations 11 and 12) and Cesa-Bianchi and Lugosi (2006, Proposition 3.3) provide a general min-max optimization framework for constructing predictions when $\ell$ is a mixable loss function. However, the optimal solution to this optimization problem depends on the specific structure of the loss function.
>
> ---
>
> **Q4:** "...there is no simulation provided..."
>
> **A4:** Dynamic regret of curved functions is a very challenging and fundamental theoretical problem in non-stationary online learning, and our primary focus has been on the theoretical aspect. Nonetheless, we are happy to include additional experiments in the revised version to further support our method.
>
> ---
>
> **Q5:** the paper should also review the dynamic regret analysis for OCO with strongly convex costs
>
> **A5:** Thanks for bringing this paper to our attention. It studies dynamic regret with switching costs, which is an interesting and complementary direction to our work. We will incorporate a discussion of this paper in the next version.
>
> ---
>
> Although the current presentation indeed has certain unclear issues, we believe the core (technical) contributions are interesting and valuable to the community. We will ensure a substantial revision to improve the clarity. Please consider updating the score if these responses have properly resolved your concerns. Thanks!

---

### Decision · Program_Chairs · 2025-05-01

**Decision:**

Accept (poster)

**Comment:**

This paper studies dynamic regret in the setting of mixable losses. The authors show that using exponential weights with fixed-share updates improves existing regret upper bounds, which previously required the stronger exp-concavity assumption and a more involved analysis.

The reviewer raised concerns about clarity, limited comparison with prior work, and the potentially prohibitive computational complexity for general losses beyond the squared loss. However, they also acknowledged the paper’s technical novelty and its significant contribution to the dynamic regret literature.

The authors are encouraged to carefully address these reviewer comments in the final revision — particularly by improving the clarity were suggested, strengthening the discussion of related work, and clarifying the applicability of the improper-to-proper reduction.